# Cost-efficient Gaussian tensor network embeddings for tensor-structured inputs

**Linjian Ma, Edgar Solomonik**
Department of Computer Science, University of Illinois at Urbana-Champaign
{lma16, solomon2}@illinois.edu

## Abstract

This work discusses tensor network embeddings, which are random matrices $(S)$ with tensor network structure. These embeddings have been used to perform dimensionality reduction of tensor network structured inputs $x$ and accelerate applications such as tensor decomposition and kernel regression. Existing works have designed embeddings for inputs $x$ with *specific* structures, such as the Kronecker product or Khatri-Rao product, such that the computational cost for calculating $Sx$ is efficient. We provide a systematic way to design tensor network embeddings consisting of Gaussian random tensors, such that for inputs with *more general* tensor network structures, both the sketch size (row size of $S$) and the sketching computational cost are low.

We analyze general tensor network embeddings that can be reduced to a sequence of sketching matrices. We provide a sufficient condition to quantify the accuracy of such embeddings and derive sketching asymptotic cost lower bounds using embeddings that satisfy this condition and have a sketch size lower than any input dimension. We then provide an algorithm to efficiently sketch input data using such embeddings. The sketch size of the embedding used in the algorithm has a linear dependence on the number of sketching dimensions of the input. Assuming tensor contractions are performed with classical dense matrix multiplication algorithms, this algorithm achieves asymptotic cost within a factor of $O(\sqrt{m})$ of our cost lower bound, where $m$ is the sketch size. Further, when each tensor in the input has a dimension that needs to be sketched, this algorithm yields the optimal sketching asymptotic cost. We apply our sketching analysis to inexact tensor decomposition optimization algorithms. We provide a sketching algorithm for CP decomposition that is asymptotically faster than existing work in multiple regimes, and show the optimality of an existing algorithm for tensor train rounding.

## 1 Introduction

Sketching techniques, which randomly project high-dimensional data onto lower dimensional spaces while still preserving relevant information in the data [43], have been widely used in numerical linear algebra, including for regression, low-rank approximation, and matrix multiplication [50]. One key step of sketching algorithms is to design an embedding matrix $S \in \mathbb{R}^{m \times n}$ with $m \ll n$, such that for any input (also called data throughout the paper) $x \in \mathbb{R}^n$, the projected vector norm is $(1 \pm \epsilon)$-close to the input vector, $\|Sx\|_2 = (1 \pm \epsilon)\|x\|_2$, with probability at least $1 - \delta$ (defined as $(\epsilon, \delta)$-accurate embedding throughout the paper), and the multiplication $Sx$ can be computationally efficient. $S$ is commonly chosen as a random matrix with each element being an i.i.d. Gaussian variable when $x$ is dense, or a random sparse matrix when $x$ is sparse, etc.

In this work, we focus on the case where $x$ has a tensor network structure. A tensor network [35] uses a set of (small) tensors, where some or all of their dimensions are contracted according to some

36th Conference on Neural Information Processing Systems (NeurIPS 2022).

pattern, to implicitly represent a tensor. Tensor network structured data is commonly seen in multiple applications, including kernel based statistical learning [38, 1, 51, 31], machine learning and data mining via tensor decomposition methods [3, 46, 22, 46], and simulation of quantum systems [49, 27, 44, 15, 16]. Commonly used embedding matrices are sub-optimal for sketching many such data. For example, consider the case where $x \in \mathbb{R}^{s^N}$ is a chain of Kronecker products, $x = x_1 \otimes \cdots \otimes x_N$, where $x_i \in \mathbb{R}^s$ for $i \in \{1, \ldots, N\}$. If $S \in \mathbb{R}^{m \times s^N}$ is a Gaussian matrix, the multiplication $Sx$ has a computational cost of $\Omega(ms^N)$, and the exponential dependence on the tensor order $N$ makes the calculation impractical when $N$ or $s$ is large.

The computational cost of the multiplication $Sx$ can be reduced when $S$ has a structure that can be easily multiplied with the target data. One example is when $S$ has a Kronecker product structure, $S = S_1 \otimes \cdots \otimes S_N$ and each $S_i \in \mathbb{R}^{m^{1/N} \times s}$. When $x = x_1 \otimes \cdots \otimes x_N$, $Sx$ can then be calculated efficiently via $(S_1 x_1) \otimes \cdots \otimes (S_N x_N)$, reducing the cost to $O(Nm^{1/N}s + m)$. Another example is when $S$ has a Khatri-Rao product structure, $S = (S_1 \odot \cdots \odot S_N)^T$ and each $S_i \in \mathbb{R}^{s \times m}$. $Sx$ can then be calculated efficiently via $(S_1^T x_1) * \cdots * (S_N^T x_N)$, where $*$ denotes the Hadamard product, which reduces the cost to $O(Nms)$. However, tensor-network-structured embedding matrices that can be easily multiplied with data may not necessarily minimize computational cost, since the sketch size sufficient for accurate embedding can also increase. For example, the sketch size necessary for both Kronecker product and Khatri-Rao product embeddings to be $(\epsilon, \delta)$-accurate is at least exponential in $N$, which is inefficient for large tensor order $N$ [1]. To find embeddings that are both accurate and computationally efficient, it is therefore of interest to investigate tensor network structures that can both yield small sketch size and be multiplied with data efficiently.

Existing works discuss tensor network embeddings with more efficient sketch size than Kronecker and Khatri-Rao product structure, such as tensor train [41] and balanced binary tree [1]. In particular, Ahle et al. [1] designed a balanced binary tree structured embedding and showed that the sketch size sufficient for $(\epsilon, \delta)$-accurate embedding can have only linear dependence on $N$. Using this embedding to sketch Kronecker product structured data yields a sketching cost that only has a polynomial dependence on both $N$ and $s$. However, for data with other tensor network structures, these embeddings may not be the most computationally efficient.

**Our contributions**  In this work, we design algorithms to efficiently sketch more general data tensor networks such that each dimension to be sketched has size lower bounded by the sketch size and is a dimension of only one tensor. One of such data tensor networks is shown in Fig. 1. In particular, we look at the following question.

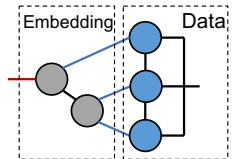

Figure 1: Illustration of target data and embedding. Blue edges have larger weights than the red edge.

*For arbitrary data with a tensor network structure of interest, can we automatically sketch the data into one tensor with Gaussian tensor network embeddings that are accurate, have low sketch size, and also minimize the sketching asymptotic computational cost?*

Different from existing works [1, 17, 26, 29] that construct the embedding based on fast sketching techniques, including Countsketch [8], Tensorsketch [37], and fast Johnson-Lindenstraus (JL) transform using fast Fourier transform [2], we discuss the case where each tensor in the embedding contains i.i.d. Gaussian random elements. Gaussian-based embeddings yield larger computational cost, but have the most efficient sketch size for both unconstrained and constrained optimization problems [9, 40]. This choice also enables us use a simple computational model to analyze the sketching cost, where tensor contractions are performed with classical dense matrix multiplication algorithms. Note that a related work [29] builds tree tensor network embeddings based on Countsketch for getting low-rank tensor network approximation of an input tensor.

While we allow for the data tensor network to be a hypergraph, we consider only graph embeddings, (detailed definition in Section 2), which include tree embeddings that have been previously studied [1, 10, 4]. Each one of these embeddings consisting of $N_E$ tensors can be reduced to a sequence of $N_E$ sketches (random sketching matrices). In Section 3, we show that if each of these sketches is $(\epsilon/\sqrt{N_E}, \delta)$-accurate, then the embedding is at least $(\epsilon, \delta)$-accurate.

In Section 4, we provide an algorithm to sketch input data with an embedding that not only satisfies the $(\epsilon, \delta)$-accurate sufficient condition, but is computationally efficient and has low sketch size.

Given a data tensor network and one data contraction tree $T_0$, this algorithm outputs a sketching contraction tree that is constrained on $T_0$. This setting is useful for application of sketching to alternating optimization in tensor-related problems, such as tensor decompositions. In alternating optimization, multiple contraction trees of the data $x$ are chosen in an alternating order to form multiple optimization subproblems, each updating part of the variables [39, 25, 28]. Designing embeddings under the constraint can help reuse contracted intermediates across subproblems.

The sketch size of the embedding used in the algorithm has a linear dependence on the number of sketching dimensions of the input. As to the sketching asymptotic computational cost, within all constrained sketching contraction trees with embeddings satisfying the $(\epsilon, \delta)$-accurate sufficient condition and only have one output sketch dimension, this algorithm achieves asymptotic cost within a factor of $O(\sqrt{m})$ of the lower bound, where $m$ is the sketch size. When the input data tensor network structure is a graph, the factor improves to $O(m^{0.375})$. In addition, when each tensor in the input data has a dimension to be sketched, such as Kronecker product input and tensor train input, this algorithm yields the optimal sketching asymptotic cost.

At the end of Section 4, we look at cases where the widely discussed tree tensor network embeddings are efficient in terms of the sketching computational cost. We show for input data graphs such that each data tensor has a dimension to be sketched and each contraction in the given data contraction tree $T_0$ contracts dimensions with size being at least the sketch size, sketching with tree embeddings can achieve the optimal asymptotic cost.

In Section 5, we apply our sketching algorithm to two applications, CANDECOMP/PARAFAC (CP) tensor decomposition [14, 13] and tensor train rounding [36]. We present a new sketching-based alternating least squares (ALS) algorithm for CP decomposition. Compared to existing sketching-based ALS algorithm, this algorithm yields better asymptotic computational cost under several regimes, such as when the CP rank is much lower than each dimension size (the length/number of elements in each dimension) of the input tensor. We also provide analysis on the recently introduced randomized tensor train rounding algorithm [10]. We show that the tensor train embedding used in that algorithm satisfies the accuracy sufficient condition in Section 3 and yields the optimal sketching asymptotic cost, implying that this is an efficient algorithm, and embeddings with other structures cannot achieve lower asymptotic cost.

## 2  Definitions

We introduce some tensor network notation here, and provide additional definitions and background in Appendix A. The structure of a tensor network can be described by an undirected hypergraph $G = (V, E, w)$, also called tensor diagram. Each hyperedge $e \in E$ may be adjacent to either one or at least two vertices, and we refer to hyperedges with a dangling end (one end not adjacent to any vertex) as uncontracted hyperedges, and those without dangling end as contracted hyperedges. We refer to the cardinality of a hyperedge as its number

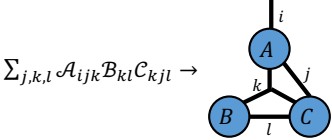

$$\Sigma_{j,k,l} \mathcal{A}_{ijk} \mathcal{B}_{kl} \mathcal{C}_{kjl} \rightarrow$$

Figure 2: An example of tensor diagram notation.

of ends. An example is shown in Fig. 2. We use $w$ to denote a function such that for each $e \in E$, $w(e) = \log(s)$ is the natural logarithm of the dimension sizes represented by hyperedge $e$. For a hyperedge set $E$, we use $w(E) = \sum_{e \in E} w(e)$ to denote the weighted sum of the hyperedge set.

A tensor network embedding is the matricization of a tensor described by a tensor network, and each embedding can be described by $S = (G_E, \bar{E})$, where $G_E = (V_E, E_E, w)$ shows the embedding graph structure and $\bar{E} \subseteq E_E$ is the edge set connecting data and the embedding. In this work we only discuss the case where $G_E$ is a graph, such that each uncontracted edge in $E_E$ is adjacent to one vertex and contracted edge in $E_E$ is adjacent to two vertices. Let $E_1 \subset E_E$ be the subset of uncontracted edges, $S$ is a matricization such that uncontracted dimensions in $\bar{E}$ are grouped into the column of the matrix, and dimensions in $E_1$ are grouped into the row. We use $N = |\bar{E}|$ to denote the order of the embedding, and $m = \exp(w(E_1))$ denotes the output sketch size. We use $G_D = (V_D, E_D, w)$ to represent the data tensor network structure, and use $G = (G_D, G_E)$ to denote the overall tensor network structure.

Within the tensor network $G = (V, E, w)$, the contraction between two tensors represented by $v_i, v_j \in V$ is denoted by $(v_i, v_j)$. The contraction between two tensors that are the contraction

outputs of $W_i \subset V$, $W_j \subset V$, respectively, is denoted by $(W_i, W_j)$. A contraction tree on the tensor network $G = (V, E, w)$ is a rooted binary tree $T_B = (V_B, E_B)$ showing how the tensor network is fully contracted. Each vertex in $V_B$ can be represented by a subset of the vertices, $W \subseteq V$, and denotes the contraction output of $W$. The two children of $W$, denoted as $W_1$ and $W_2$, must satisfy $W_1 \cup W_2 = W$. Each leaf vertex must have $|W| = 1$, and the root vertex is represented by $V$. Any topological sort of the contraction tree represents a contraction path (order) of the tensor network.

# 3   Sufficient condition for accurate embedding

We consider the scenario where the data tensor networks have a general hypergraph structure, while the embeddings have a graph structure, thus some embeddings, such as those with a Khatri-Rao product structure [41, 9], are not considered in this work. Such embeddings can be linearized to a sequence of sketches. Let $N_E = |V_E|$ denote the number of vertices in the embedding, in each linearization, each vertex is given an unique index $i \in [N_E]$[1] and denoted $v_i$. The $i$th

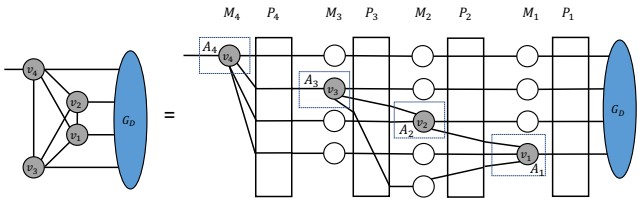

Figure 3: Illustration of embedding linearization. Each gray vertex denotes a tensor of the embedding, each white vertex denotes an identity matrix, and each white box denotes a permutation matrix.

tensor is denoted by $\mathcal{A}_i$, and $A_i$ denotes its matricization where we combine all uncontracted dimensions and contracted dimensions connected to $\mathcal{A}_j$ with $j > i$ into the row, and other dimensions into the column. The embedding can then be represented as a chain of multiplications, $S = M_{N_E} P_{N_E} \cdots M_1 P_1$, where $M_i$ is the Kronecker product of identity matrices with $A_i$ for $i \in [N_E]$, and $P_i$ is a permutation matrix. We illustrate the linearization in Fig. 3 using a fully connected tensor network embedding. We show in Theorem 3.1 a sufficient condition for embeddings to be $(\epsilon, \delta)$-accurate.

**Theorem 3.1** (($\epsilon, \delta$)-accurate sufficient condition). *Consider a Gaussian tensor network embedding where there exists a linearization such that each $A_i$ for $i \in [N_E]$ has row size $\Omega(N_E \log(1/\delta)/\epsilon^2)$. Then the tensor network embedding is $(\epsilon, \delta)$-accurate.*

*Proof.* Based on the composition rules of JL moment [18, 19] in Lemma A.2 and Lemma A.3 in the appendix, in the linearization all $M_i P_i$ satisfy the strong $\left(\frac{\epsilon}{L\sqrt{2N}}, \delta\right)$-JL moment property so $S$ satisfies the strong $(\epsilon, \delta)$-JL moment property. This implies the embedding is $(\epsilon, \delta)$-accurate. $\square$

Theorem 3.1 is a sufficient (but not necessary) condition for constructing $(\epsilon, \delta)$-accurate embedding. It also implies that specific tree embeddings are $(\epsilon, \delta)$-accurate, as we show below.

**Corollary 3.2.** *Consider a Gaussian embedding containing a tree tensor network structure, where there is only one output sketch dimension with size $m = \Theta(N_E \log(1/\delta)/\epsilon^2)$, and each dimension within the embedding has size $m$. Then the embedding is $(\epsilon, \delta)$-accurate.*

*Proof.* Consider the linearization such that vertices are labelled based on the reversed ordering of a breath-first search from the vertex adjacent to the edge associated with the output sketch dimension. Each $A_i$ has row size $m = \Theta(N_E \log(1/\delta)/\epsilon^2)$ thus the embedding satisfies Theorem 3.1. $\square$

One special case of Corollary 3.2 is the tensor train [36] (also called matrix product states (MPS) [45]) embedding, where the embedding tensor network has a 1D structure along with an output dimension adjacent to one of the endpoint tensors. Tensor train is widely used to efficiently represent high dimensional tensors in multiple applications, including numerical PDEs [12, 42], quantum physics [44], high-dimensional data analysis [20, 21] and machine learning [6, 47, 33]. Since the tensor train embedding contains $N$ vertices, Corollary 3.2 directly implies that a sketch size of $m = \Theta(N \log(1/\delta)/\epsilon^2)$ is sufficient for the MPS embedding to be $(\epsilon, \delta)$-accurate. This embedding has already been used in applications including tensor train rounding [10] and low rank approximation of matrix product operators [4].

---

[1]Throughout the paper we use $[N]$ to denote $\{1, \ldots, N\}$.

Note that the tensor train embedding introduced in this work and [10] adds an output sketch dimension to the standard tensor train, and restricts the tensor train rank to be the sketch size $m$. This is different from the recent work by Rakhshan and Rabusseau [41], where they construct an embedding consisting of $m$ independent tensor trains, each one with a tensor train rank of $R$. A sketch size upper bound of $m = \Theta\left(1/\epsilon^2 \cdot (1 + 2/R)^N \log^{2N}(1/\delta)\right)$ is derived for that embedding to be $(\epsilon, \delta)$-accurate. However, this bound has an exponential dependence on $N$.

## 4  A sketching algorithm with efficient computational cost and sketch size

We find Gaussian tensor network embeddings $G_E$ that both have efficient sketch size and yield efficient computational cost. We are given a specific data tensor network $G_D$ that implicitly represents a matrix $M \in \mathbb{R}^{s_1 s_2 \cdots s_N \times t}$, and want to sketch the row dimension of the matrix. We assume that size of each dimension to be sketched, $s_i$ for $i \in [N]$, is greater than the sketch size $m$, and each one of these dimensions is adjacent to only one tensor. The goal is to find a Gaussian embedding $G_E$ satisfying the following properties.

- $G_E \in \mathcal{G}^{(\epsilon,\delta)}$, where $\mathcal{G}^{(\epsilon,\delta)}$ contains all embeddings not only satisfying the $(\epsilon, \delta)$-accurate sufficient condition in Theorem 3.1, but also only have one output sketch dimension ($|E_1| = 1$) with size $m = \Theta\left(N_E \log(1/\delta)/\epsilon^2\right)$. This guarantees that the embedding is accurate and the output sketch size is linear w.r.t. the number of vertices in $G_E$. Note that although the data can be a hypergraph, the embeddings considered in $\mathcal{G}^{(\epsilon,\delta)}$ are defined on graphs.
- To fully contract the tensor network system $(G_D, G_E)$, this embedding yields a contraction tree with the *optimal asymptotic contraction cost* under a fixed data contraction tree. The data contraction tree constraint is useful for application of sketching to alternating optimization algorithms, as we will discuss in Section 5. This can be written as an optimization problem below,

$$\min_{G_E} \min_{T_B} C_a(T_B(G_D, G_E)), \quad \text{s.t. } G_E \in \mathcal{G}^{(\epsilon,\delta)}, \quad T_0(G_D) \subset T_B(G_D, G_E), \quad (4.1)$$

where $T_B(G_D, G_E)$ denotes a contraction tree of the tensor network $(G_D, G_E)$, $C_a$ denotes the asymptotic computational cost, and $T_0(G_D) \subset T_B(G_D, G_E)$ means the contraction tree $T_B$ is constrained on $T_0$ (the detailed definition and a simple example are shown in Definition 1 and Appendix B.1, respectively).

**Definition 1** (Constrained contraction tree). Given $G = (G_E, G_D)$ and a contraction tree $T_0$ of $G_D$, the contraction tree $T_B$ for $G$ is constrained on $T_0$ if for each contraction $(A, B) \in T_0$, there must exist one contraction $(\hat{A}, \hat{B}) \in T_B$, such that $\hat{A} \cap V_D = A$ and $\hat{B} \cap V_D = B$.

**Algorithm**  We propose an algorithm to sketch tensor network data with an embedding containing two parts, a Kronecker product embedding and an embedding containing a binary tree of small tensor networks. The embedding is illustrated in Fig. 4. The Kronecker product embedding consists of $N$ Gaussian random matrices and is used to reduce the weight of each edge in $\bar{E}$, the set of edges to be sketched. The binary tree structured embedding consists of $N - 1$ *small tensor networks*, each represented by one binary tree vertex. Each small tensor network is used to effectively sketch the contraction of pairs of tensors adjacent to edges in $\bar{E}$. The embedding with a binary tree structure may not be a binary tree tensor network, since each tree vertex is not restricted to represent one tensor. The binary tree is chosen to be consistent with the dimension tree of the data contraction tree $T_0$, which is a directed binary tree showing the way edges in $\bar{E}$ are merged onto the same tensor in $T_0$. The detailed definition of dimension tree is in Appendix B.1.

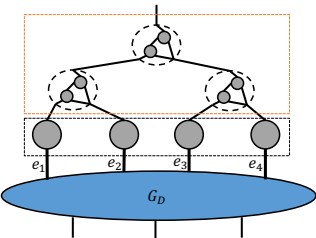

Figure 4: Illustration of the embedding. The black box includes the Kronecker product embedding and the orange box includes the embedding containing a binary tree of small tensor networks.

We first introduce some notation before presenting the algorithm.
Consider a given input data tensor network $G_D = (V_D, E_D, w)$ and its given data contraction tree, $T_0$. Below we let $\bar{E} = \{e_1, e_2, \ldots, e_N\}$ to denote the edges to be sketched. Let $N_D = |V_D|$. Based on the definition we have $N \leq N_D$ and $T_0$ contains $N_D - 1$ contractions. Let one contraction path

of $T_0$, which is a topological sort of the contractions in $T_0$, be expressed as

$$\{(U_1, V_1), \ldots, (U_{N_D-1}, V_{N_D-1})\}, \tag{4.2}$$

where $(U_i, V_i)$ represents the contraction of two intermediate tensors represented by two subset of vertices $U_i, V_i \subset V_D$. The $N_D - 1$ contractions can be categorized into $N + 2$ sets, $\mathcal{D}(e_1), \ldots, \mathcal{D}(e_N), \mathcal{S}, \mathcal{I}$, as follows, and these sets are illustrated with an example in Appendix C.

- Consider contractions $(U_i, V_i)$ such that both $U_i$ and $V_i$ are adjacent to edges in $\bar{E}$. $\mathcal{S}$ contains all contractions with this property.
- Consider contractions $(U_i, V_i)$ such that the only edge in $\bar{E}$ that is adjacent to the contraction output is $e_j$, $\bar{E}(U_i \cup V_i) = \{e_j\}$. We let $\mathcal{D}(e_j)$ contains contractions with this property. When $\mathcal{D}(e_j)$ is not empty, we let $X(e_j) \subset V$ represent the sub network contracted by $\mathcal{D}(e_j)$. When $\mathcal{D}(e_j)$ is empty, we let $X(e_j) = v_j$, where $v_j$ is the vertex in the data graph adjacent to $e_j$.
- The remaining contractions in the contraction tree include $(U_i, V_i)$ such that both $U_i$ and $V_i$ are not adjacent to $\bar{E}$, and contractions where $U_i$ or $V_i$ is adjacent to at least two edges in $\bar{E}$, and the other one is not adjacent to any edge in $\bar{E}$. We let $\mathcal{I}$ contain these contractions.

The sketching algorithm is shown in Algorithm 1, and the details are as follows,

- One matrix in the Kronecker product embedding is used to sketch the sub data network $X(e_j)$, which guarantees that two sketch dimensions to be merged onto one tensor will both have size $\Theta(N \log(1/\delta)/\epsilon^2)$. For the case where $\mathcal{D}(e_j) = \emptyset$, we directly sketch $X(e_j) = v_j$ using an embedding matrix. For the case where $\mathcal{D}(e_j) \neq \emptyset$, we select $k(e_j) \in \mathcal{D}(e_j)$ and apply the sketching matrix during the contraction $(U_{k(e_j)}, V_{k(e_j)})$. The value of $k(e_j)$ is selected via an exhaustive search over all $|\mathcal{D}(e_j)|$ contractions, so that sketching $X(e_j)$ has the lowest asymptotic cost.
- One small tensor network (denoted as $Z_i$) represented by a binary tree vertex in the binary tree structured embedding is used to sketch the contraction $(U_i, V_i)$ when $i \in \mathcal{S}$, which means that both $U_i$ and $V_i$ are adjacent to $\bar{E}$. Let $\hat{U}_i, \hat{V}_i$ denote the sketched $U_i$ and $V_i$ formed in previous contractions

---

**Algorithm 1** Sketching algorithm

1: **Input:** Input data tensor network $G_D$, data contraction tree $T_0$ expressed in (4.2)
2: **for** each $e_i \in \bar{E}$ **do**
3:     // *Sketch with Kronecker product embedding*
4:     $W \leftarrow$ contract and sketch $X(e_j)$
5:     Replace the contraction output of $X(e_j)$ by $W$ in $T_0$
6: **end for**
7: **for** each contraction $(U_i, V_i)$ in $\mathcal{S} \cup \mathcal{I}$ **do**
8:     **if** $i \in \mathcal{S}$ **then**
9:       // *Sketch with binary tree embedding*
10:       $W_i \leftarrow$ contract and sketch $(U_i, V_i)$ (detailed in Appendix C.1)
11:     **else**
12:       $W_i \leftarrow \text{contract}(U_i, V_i)$
13:     **end if**
14:     Replace the contraction output of $(U_i, V_i)$ by $W_i$ in $T_0$
15: **end for**
16: **return** $W_{N_D-1}$

---

in the sketching contraction tree $T_B$, such that $\hat{U}_i \cap V_D = U_i$ and $\hat{V}_i \cap V_D = V_i$, the structure of $Z_i$ is determined so that the asymptotic cost to sketch $(\hat{U}_i, \hat{V}_i)$ is minimized under the constraint that $Z_i$ is in $\mathcal{G}^{(\epsilon/\sqrt{N}, \delta)}$, so that it satisfies the $(\epsilon/\sqrt{N}, \delta)$-accurate sufficient condition and only has one output dimension. In Appendix C.1, we provide an algorithm to construct $Z_i$ containing 2 tensors, so that the output sketch size of $Z_i$ is $\Theta(N \log(1/\delta)/\epsilon^2)$.

The total computational cost of Algorithm 1 consists of three components: the cost of determining the embedding structure, the cost of determining the sketching contraction tree, and the cost of sketching. The first two components are $O(N)$ and are therefore negligible in comparison to the cost of sketching.

**Analysis of the algorithm** The embedding constructed during Algorithm 1 contains $\Theta(N)$ vertices, and the output sketch size is $m = \Theta(N \log(1/\delta)/\epsilon^2)$. Therefore, the sketching result both has low sketch size and is $(\epsilon, \delta)$-accurate. Below we discuss the optimality of Algorithm 1 in terms of the sketching asymptotic computational cost. We first discuss the case when each vertex in the data tensor network is adjacent to an edge in $\bar{E}$.

**Theorem 4.1.** *For data tensor networks where each vertex is adjacent to an edge in $\bar{E}$, the asymptotic cost of Algorithm 1 is optimal w.r.t. the optimization problem in (4.1).*

We show the detailed proof of the theorem above in Appendix D.1. Therefore, Algorithm 1 is efficient in sketching multiple widely used tensor network data, including tensor train, Kronecker product, and Khatri-Rao product. As we will discuss in Section 5, Algorithm 1 can be used to design efficient sketching-based ALS algorithm for CP tensor decomposition.

Note that the embedding in Algorithm 1 may not be a tree embedding. As we will show in Section 6, for cases including sketching a Kronecker product data, Algorithm 1 is more efficient than sketching with tree embeddings. On the other hand, for some data tensor networks, sketching with a tree embedding also yields the optimal asymptotic cost, which we will show in Theorem 4.3.

For general input data where each data vertex may not adjacent to an edge in $\bar{E}$, Algorithm 1 may not yield the optimal sketching asymptotic cost, but is within a factor of at most $O(\sqrt{m})$ from the cost lower bound. Below we show the theorem, and the detailed proof is in Appendix D.2.

**Theorem 4.2.** *For any data tensor network $G_D$, the asymptotic cost of Algorithm 1 (denoted as c) satisfy $c = O(\sqrt{m} \cdot c_{opt})$, where $c_{opt}$ is the optimal asymptotic computational cost for the optimization problem* (4.1) *and $m = \Theta(N \log(1/\delta)/\epsilon^2)$. When $G_D$ is a graph, $c = O\left(m^{0.375} \cdot c_{opt}\right)$.*

**Efficiency of tree tensor network embedding**    We discuss cases where tree tensor network embeddings can be optimal w.r.t. the optimization problem in (4.1). Tree embeddings, in particular the tensor train embedding, have been widely discussed and used in prior work [41, 10, 4]. We design an algorithm to sketch with tree embeddings. The algorithm is similar to Algorithm 1, and the only difference is that for each contraction $(U_i, V_i)$ with $i \in \mathcal{S}$, such that both $U_i$ and $V_i$ are adjacent to edges in $\bar{E}$, we sketch it with one tensor rather than a small network. Below, we present the optimality of the algorithm in terms of sketching asymptotic cost.

**Theorem 4.3.** *Consider $G_D$ with each vertex adjacent to an edge to be sketched and its given contraction tree $T_0$. If each contraction in $T_0$ contracts dimensions with size being at least the sketch size, then sketching with tree embedding would yield the optimal asymptotic cost for* (4.1).

We present the proof of Theorem 4.3 in Appendix E. As we will show in Section 6, for tensor network data with relatively large contracted dimension sizes such that the condition in Theorem 4.3 is satisfied, sketching with tree embedding yields a similar performance as Algorithm 1. However, for data where the condition in Theorem 4.3 is not satisfied, Algorithm 1 is more efficient. For example, when the data is a vector with a Kronecker product structure, sketching with Algorithm 1 yields a cost of $\Theta(\sum_{j=1}^{N} s_j m + N m^{2.5})$ and sketching with a tree embedding yields a cost of $\Theta(\sum_{j=1}^{N} s_j m + N m^3)$. We present the detailed analysis in Appendix C.2 and Appendix E.

## 5    Applications

**Alternating least squares for CP decomposition**    On top of Algorithm 1, we propose a new sketching-based ALS algorithm for CP tensor decomposition. Throughout analysis we assume the input tensor is dense, and has order $N$ and size $s \times \cdots \times s$, and the CP rank is $R$. The goal of CP decomposition is to minimize the objective function, $f(A_1, \ldots, A_N) = \left\| \mathcal{X} - \sum_{r=1}^{R} A_1(:, r) \circ \cdots \circ A_N(:, r) \right\|_F^2$, where $A_i \in \mathbb{R}^{s \times R}$ for $i \in [N]$ are called factor matrices, and $\mathcal{X}$ denotes the input tensor. In each iteration of ALS, $N$ subproblems are solved sequentially, and the $i$th subproblem can be formulated as $A_i = \arg\min_A \left\| L_i A^T - R_i \right\|_F^2$, where $L_i = A_1 \odot \cdots \odot A_{i-1} \odot A_{i+1} \odot \cdots \odot A_N$ consists of a chain of Khatri-Rao products, and $R_i = X_{(i)}^T$ is the transpose of $i$th matricization of $\mathcal{X}$.

Multiple sketching-based randomized algorithms are proposed to accelerate each subproblem in CP-ALS [5, 24, 30]. The sketched problem can be formulated as $A_i = \underset{A}{\arg\min} \left\| S_i L_i A^T - S_i R_i \right\|_F^2$, where $S_i$ is an embedding. The goal is to design $S_i$ such that the sketched subproblem can be solved efficiently and accurately. In Table 1, we summarize two state-of-the-art sketching methods for CP-ALS. Larsen and Kolda [24] propose a method that sketches the subproblem based on (approximate) leverage score sampling (LSS), but both the per-iteration computational cost and the sketch size sufficient for $(\epsilon, \delta)$-accurate solution has an exponential dependence on $N$, which is inefficient for decomposing high order tensors. Malik [30] proposes a method called recursive leverage score

sampling for CP-ALS, where the embedding contains two parts, $S_i = S_{i,1}S_{i,2}$, and $S_{i,2}$ is an embedding with a binary tree structure proposed in [1] with sketch size $\Theta(NR^2/\delta)$, and $S_{i,1}$ performs approximate leverage score sampling on $S_{i,2}L_i$ with sketch size $\tilde{\Theta}(NR/\epsilon^2)$. This sketching method has a better dependence on $R$ in terms of per-iteration cost. For both algorithms, the preparation cost shown in Table 1 denotes the cost to go over all elements in the tensor and initialize factor matrices using randomized range finder. As is shown in [24, 26], randomized range finder based initialization is critical for achieving accurate CP decomposition with sampling-based sketched ALS.

| CP-ALS algorithm | Per-iteration cost | Sketch size ($m$) | Prep cost |
|---|---|---|---|
| Standard ALS | $\Theta(s^N R)$ | / | / |
| LSS [24] | $\tilde{\Theta}(N(R^{N+1} + sR^N)/\epsilon^2)$ | $\tilde{\Theta}(R^{N-1}/\epsilon^2)$ | $\Theta(s^N)$ |
| Recursive LSS [30] | $\tilde{\Theta}(N^2(R^4 + NsR^3/\epsilon)/\delta)$ | $\Theta(NR^2/\delta)$ and $\tilde{\Theta}(R/(\epsilon\delta))$ | $\Theta(s^N)$ |
| Algorithm 1 | $\tilde{\Theta}(N^2(N^{1.5}R^{3.5}/\epsilon^3 + sR^2)/\epsilon^2)$ | $\tilde{\Theta}(NR/\epsilon^2)$ | $\Theta(s^N m)$ |

Table 1: Comparison of asymptotic algorithmic complexity between standard CP-ALS, CP-ALS with leverage score sampling (LSS), CP-ALS with recursive leverage score sampling (recursive LSS), and sketching CP-ALS with Algorithm 1. The third column shows the sketch size sufficient for the sketched linear least squares to be $(1 + \epsilon)$-accurate with probability at least $1 - \delta$. By using $\tilde{\Theta}$, we neglect logarithmic factors, including $\log(R)$ and $\log(1/\delta)$.

We propose a new sketching algorithm for CP-ALS based on Algorithm 1. Each $S_i$ is generated on top of the data tensor network $L_i$ and its given data contraction tree $T_i$, with the sketch size being $m = \Theta(NR\log(1/\delta)/\epsilon^2) = \tilde{\Theta}(NR/\epsilon^2)$. The contraction trees $T_i$ for $i \in [N]$ are chosen in a fixed alternating order, such that the resulting embeddings $S_i$ for $i \in [N]$ have common parts and allow reusing contraction intermediates. We leave the detailed analysis in Appendix F.2.

The ALS per-iteration cost is $\Theta(N(m^{2.5}R + smR)) = \tilde{\Theta}(N^2(N^{1.5}R^{3.5}/\epsilon^3 + sR^2)/\epsilon^2)$. We present the detailed sketching algorithm and its cost analysis in Appendix F.3. When performing a low-rank CP decomposition with $s \gg R^{1.5}$ and $\epsilon$ is not too small so that $\epsilon = \Theta(1)$[2], the per-iteration cost is dominated by the term $\tilde{\Theta}(N^2sR^2/\epsilon^2)$, which is $\tilde{\Theta}(NR\epsilon/\delta) = \Omega(NR)$ times better than the per-iteration cost of the recursive LSS algorithm. For another case of a high-rank CP decomposition with $R \gg s$, which happens when one wants a high-accuracy CP decomposition of high order tensors, the per-iteration cost of our sketched CP-ALS algorithm is dominated by the term $\tilde{\Theta}(N^{3.5}R^{3.5}/\epsilon^5)$, and the cost ratio between this algorithm and the recursive LSS algorithm is $\tilde{\Theta}(N^{1.5}\delta/(\epsilon^5 R^{0.5}))$. For this case, our algorithm is only preferable when $N^{1.5}\delta/\epsilon^5$ is not too large compared to $R^{0.5}$.

Although our proposed sketching algorithm yields better per-iteration asymptotic cost in multiple regimes compared to existing leverage score based sketching algorithms, some preparation computations are needed to sketch right-hand-sides $S_iR_i$ for $i \in [N]$ before ALS iterations, and this cost is non-negligible. On the other hand, this algorithm has better parallelism, since it involves a sequence of matrix multiplications rather than sampling the matrix. We leave the detailed experimental comparison of computational efficiency of different sketching techniques for future work.

**Tensor train rounding** Given a tensor train, *tensor train rounding* finds a tensor train with a lower rank to approximate the original representation. Throughout analysis we assume the tensor train has order $N$ with the output dimension sizes equal $s$, the tensor train rank is $R < s$, and the goal is to round the rank to $r < R$. The standard tensor train rounding algorithm [36] consists of a right-to-left sweep of QR decompositions of the input tensor train (also called orthogonalization), and another left-to-right truncated singular value decompositions (SVD) sweep to perform rank reduction. The orthogonalization step is the bottleneck of the rounding algorithm, and costs $\Theta(NsR^3)$. Recently, [10] has introduced a randomized rounding algorithm called "*Randomize-then-Orthogonalize*". Let $X$ denote a matricization of the tensor train data with all except one dimension at the end grouped into the row, the algorithm first sketches $X$ with a tensor train embedding $S$, then performs a sequence of truncated SVDs on top of $SX$. The sketch size $m$ of $S$ is $r$ plus some constant, and is assumed to be smaller than $R$. The bottleneck is to compute $SX$, which costs $\Theta(NsR^2m)$.

---

[2]As is shown in [26], in practice, setting $\epsilon$ to be 0.1-0.2 will result in accurate sketched least squares with relative residual norm error less than 0.05.

We can recast the problem as finding an embedding satisfying the linearization sufficient condition with sketch size $m$, such that the asymptotic cost of computing $SX$ is optimal given the data contraction tree that contracts the tensor train from one end to another. Our analysis (detailed in Appendix G) shows that the sketching cost for the problem is lower bounded by $\Omega(NsR^2m)$, thus the sketching algorithm in [10] attains the asymptotic cost lower bound and is efficient. Note that sketching with Algorithm 1 yields the same asymptotic cost, despite using a different embedding.

## 6 Experiments

We conduct multiple experiments to demonstrate the efficacy of our proposed embeddings. Below we first justify the theoretical analysis in Theorem 4.1 and Theorem 4.3 via testing the sketching performance on tensor train inputs and Kronecker product inputs. We then perform experiments to demonstrate that the accuracy of our proposed sketching algorithms is comparable to that of state-of-the-art sketching techniques for CP decomposition and tensor train rounding. Our experiments are carried out on an Intel Core i7 2.9 GHz Quad-Core machine using NumPy [34] routines in Python.

**Sketching tensor train and Kronecker product inputs**   We compare the performance of general tensor network embedding used in Algorithm 1 (called TN embedding), tree embedding discussed in Theorem 4.3, and the baseline, tensor train embedding [10], in sketching tensor train input data in Fig. 5. The input tensor train data has order 6, and the dimension size is 500.

We test the sketching performance under different tensor train ranks. For a given rank, we randomly generate 25 different inputs, with each element in each tensor being an i.i.d. variable uniformly distributed within $[0, 1]$. Additional experiments with Gaussian-distributed input tensor train data are presented in Appendix H. For each input $x$ and a specific embedding structure, we calculate the relative sketching error twice under different sketch sizes, and record the smallest sketch size such that both of its relative sketching errors

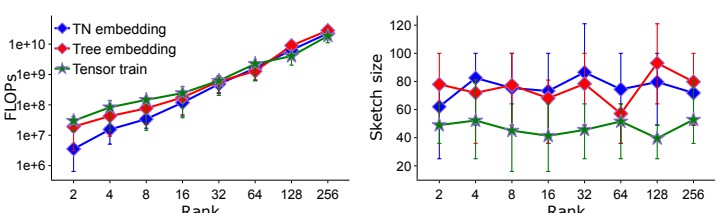

Figure 5: Results for sketching tensor train inputs. Each point denotes the mean value across 25 experiments, and each error bar shows the 25th-75th quartiles.

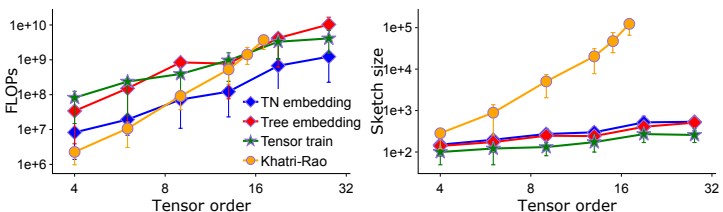

Figure 6: Results for sketching Kronecker product inputs.

are within 0.2, $\frac{\|Sx\|_2}{\|x\|_2} \leq 0.2$. We also calculate the number of floating point operations (FLOPs) for computing $Sx$ under the smallest sketch size based on the classical dense matrix multiplication algorithm. As can be seen, tree and tensor train embeddings are as efficient as TN embedding in terms of number of FLOPs under relatively high tensor train rank (when rank is at least 32), but are less efficient than TN embedding when the tensor train rank is lower than 32. The results are consistent with the theoretical analysis in Theorem 4.3, which shows that tree embeddings yield the optimal asymptotic cost when the input tensor train rank is at least the output sketch size, but the asymptotic cost is not optimal when the tensor train rank is low.

We also compare the performance of TN, tree, and two baselines proposed in [41], tensor train and Khatri-Rao product embeddings, in sketching Kronecker product inputs in Fig. 6. Each dimension size of the Kronecker product input is fixed to be 1000, and we test the sketching performance under different tensor orders. For each input $x$ and a specific embedding structure, we record the smallest sketch size such that its relative sketching error is within 0.1. As can be seen, compared to Khatri-Rao product embedding, the sketch size of TN, tree and tensor train embeddings all increase slowly with the increase of tensor order, consistent with the theoretical analysis that these embeddings have efficient sketch size. In addition, the cost in FLOPs of TN embedding is smaller than tree and tensor

train embeddings. This is consistent with the analysis in Theorem 4.1 and its following discussions, showing that TN embedding yields the optimal asymptotic cost for Kronecker product inputs, but tree and tensor train embeddings do not.

**CP decomposition and tensor train rounding**   We perform experiments to demonstrate that the accuracy of our proposed sketching methods for CP-ALS and tensor train rounding is comparable to that of state-of-the-art sketching techniques. For both applications, we evaluate accuracy based on the final fitness $f$ for each algorithm, defined as $f = 1 - \frac{\|\mathcal{T} - \widetilde{\mathcal{T}}\|_F}{\|\mathcal{T}\|_F}$, where $\mathcal{T}$ is the input tensor and $\widetilde{\mathcal{T}}$ is the reconstructed low-rank tensor.

For CP-ALS, we conduct experiments on a Time-Lapse hyperspectral radiance image [32], which is a 3-D tensor with dimensions of $1024 \times 1344 \times 33$. This input data is used to demonstrate the applicability of our method in real-world scenarios. Standard CP-ALS, sketched CP-ALS using Algorithm 1, and sketched CP-ALS with approximate leverage score sampling (LSS) [24] are compared. The output CP decomposition fitness under varying CP ranks and sketch sizes is shown in Table 2. As can be seen, sketching with Algorithm 1 yields comparable fitness with the algorithm that sketches with approximate leverage score sampling. Note that the computational cost of Algorithm 1 is lower that LSS especially when the CP rank is low and the tensor dimension is large, as stated in Table 1.

| CP rank | 2 | 5 | 10 |
|---|---|---|---|
| Sketch size | 25 | 64 | 100 |
| CP-ALS | 0.737 | 0.804 | 0.838 |
| LSS [24] | 0.739 | 0.773 | 0.789 |
| Algorithm 1 | 0.737 | 0.770 | 0.801 |

Table 2: Comparison of the final fitness of different CP decomposition algorithms under different CP ranks and sketch sizes. 10 ALS iterations are performed for all algorithms before the final fitness are calculated.

| TT rounding rank | 1 | 4 | 11 | 20 |
|---|---|---|---|---|
| Sketch size | 4 | 9 | 16 | 25 |
| TT-SVD [36] | 0.734 | 0.862 | 0.944 | 0.981 |
| TT embedding [10] | 0.573 | 0.757 | 0.882 | 0.951 |
| Algorithm 1 | 0.527 | 0.761 | 0.866 | 0.948 |

Table 3: Comparison of the final fitness of different tensor train rounding algorithms under different tensor train rounding ranks and sketch sizes.

We use 9 images from the Time-Lapse hyperspectral radiance image dataset for tensor train rounding, and reshape the input data to an order 6 tensor with size $9 \times 32 \times 32 \times 28 \times 48 \times 33$. We use the TensorLy [23] library to truncate the input tensor to a tensor train with a rank of 30. On top of this tensor train, we evaluate the accuracy of various approaches, including tensor train SVD [36], randomized algorithm using tensor train embedding [10], and randomized algorithm using Algorithm 1. The fitness of the truncated tensor trains are displayed in Table 3 for a variety of rounding rank thresholds and sketch sizes. As can be seen, sketching with Algorithm 1 has comparable accuracy with the baseline algorithm (sketching with tensor train embedding). In addition, both sketching algorithms also have similar complexity as is analyzed in Section 5.

# 7   Conclusions

We provide detailed analysis of general tensor network embeddings. For input data such that each dimension to be sketched has size greater than the sketch size, we provide an algorithm to efficiently sketch such data using Gaussian embeddings that can be linearized into a sequence of sketching matrices and have low sketch size. Our sketching method is then used to design state-of-the-art sketching algorithms for CP tensor decomposition and tensor train rounding. We leave the analysis for more general embeddings for future work, including those with each tensor representing fast sketching techniques, such as Countsketch and fast JL transform using fast Fourier transform, and those containing structures cannot be linearized, such as Khatri-Rao product embedding. It would also be of interest to look at other tensor-related applications that could benefit from tensor network embedding, including tensor ring decomposition and simulation of quantum circuits. We also leave the high-performance implementation of the algorithm for general tensor networks as future work.

## Acknowledgments

This work is supported by the US NSF OAC via award No. 1942995.

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
