# A Background

## A.1 Tensor algebra and tensor diagram notation

Our analysis makes use of tensor algebra for tensor operations [22]. Vectors are denoted with lowercase Roman letters (e.g., $v$), matrices are denoted with uppercase Roman letters (e.g., $M$), and tensors are denoted with calligraphic font (e.g., $\mathcal{T}$). An order $N$ tensor corresponds to an $N$-dimensional array. For an order $N$ tensor $\mathcal{T} \in \mathbb{R}^{s_1 \times \cdots \times s_N}$, the size of $i$th dimension is $s_i$. The $i$th column of the matrix $M$ is denoted by $M(:, i)$, and the $i$th row is denoted by $M(i, :)$. Subscripts are used to label different vectors, matrices and tensors (e.g. $\mathcal{T}_1$ and $\mathcal{T}_2$ are unrelated tensors). The Kronecker product of two vectors/matrices is denoted with $\otimes$, and the outer product of two or more vectors is denoted with $\circ$. For matrices $A \in \mathbb{R}^{m \times k}$ and $B \in \mathbb{R}^{n \times k}$, their Khatri-Rao product results in a matrix of size $(mn) \times k$ defined by $A \odot B = [A(:,1) \otimes B(:,1), \ldots, A(:,k) \otimes B(:,k)]$. Matricization is the process of unfolding a tensor into a matrix. The dimension-$n$ matricized version of $\mathcal{T}$ is denoted by $T_{(n)} \in \mathbb{R}^{s_n \times K}$ where $K = \prod_{m=1, m \neq n}^{N} s_m$.

We introduce the graph representation for tensors, which is also called tensor diagram [7]. A tensor is represented by a vertex with hyperedges adjacent to it, each corresponding to a tensor dimension. A matrix $M$ and an order four tensor $\mathcal{T}$ are represented as follows,

$$M \implies -\!\bigcirc\!- \qquad \mathcal{T} \implies \bigcirc .$$

The Kronecker product of two matrices $A$ and $B$ can be expressed as

$$\left( A \quad B \right) = \left( A \otimes B \right) .$$

Connecting two edges means two tensor dimensions are contracted or summed over. One example is shown in Fig. 7.

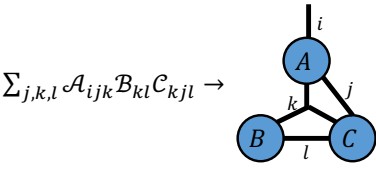

$$\sum_{j,k,l} \mathcal{A}_{ijk} \mathcal{B}_{kl} \mathcal{C}_{kjl} \rightarrow$$

Figure 7: An example of tensor diagram notation.

## A.2 Background on sketching

In this section, we introduce definitions for sketching used throughout the paper.

**Definition 2** (Gaussian embedding). A matrix $S = \frac{1}{\sqrt{m}} M \in \mathbb{R}^{m \times n}$ is a Gaussian embedding if each element of $M$ is a normalized Gaussian random variable, $M(i, j) \sim N(0, 1)$.

One key property we would like the tensor network embedding to satisfy is the $(\epsilon, \delta)$-accurate property. To achieve this, one central property each tensor in the tensor network embedding needs to satisfy is the Johnson-Lindenstrauss (JL) moment property. The JL moment property captures a bound on the moments of the difference between the vector Euclidean norm and the norm after sketching. We introduce both definitions below.

**Definition 3** (($\epsilon$, $\delta$)-accurate embedding). A random matrix $S \in \mathbb{R}^{m \times n}$ has the $(\epsilon, \delta)$-accurate embedding property if for every $x \in \mathbb{R}^n$ with $\|x\|_2 = 1$,

$$\Pr_{S} \left( \left| \|Sx\|_2^2 - 1 \right| > \epsilon \right) < \delta.$$

**Definition 4** (($\epsilon$, $\delta$, $p$)-JL moment [18, 19]). A random matrix $S \in \mathbb{R}^{m \times n}$ has the $(\epsilon, \delta, p)$-JL moment property if for every $x \in \mathbb{R}^n$ with $\|x\|_2 = 1$,

$$\underset{S}{\mathbb{E}} \left| \|Sx\|_2^2 - 1 \right|^p < \epsilon^p \delta \quad \text{and} \quad \mathbb{E}\left[ \|Sx\|_2^2 \right] = 1.$$

**Definition 5** (Strong $(\epsilon, \delta)$-JL moment [18, 1]). A random matrix $S \in \mathbb{R}^{m \times n}$ has the strong $(\epsilon, \delta)$-JL moment property if for every $x \in \mathbb{R}^n$ with $\|x\|_2 = 1$, and every integer $p \in [2, \log(1/\delta)]$,

$$\underset{S}{\mathbb{E}} \left| \|Sx\|_2^2 - 1 \right|^p < \left( \frac{\epsilon}{e} \right)^p \left( \frac{p}{\log(1/\delta)} \right)^{p/2} \tag{A.1}$$

and $\mathbb{E}\left[ \|Sx\|_2^2 \right] = 1$.

Note that the strong $(\epsilon, \delta)$-JL moment property directly reveals the $(\epsilon, \delta, \log(1/\delta))$-JL moment property, since letting $p = \log(1/\delta)$, (A.1) becomes

$$\underset{S}{\mathbb{E}} \left| \|Sx\|_2^2 - 1 \right|^{\log(1/\delta)} < \left( \frac{\epsilon}{e} \right)^{\log(1/\delta)} = \epsilon^p \delta.$$

Both the strong $(\epsilon, \delta)$-JL moment property and the $(\epsilon, \delta, p)$-JL moment property directly imply $(\epsilon, \delta)$-accurate embedding via Markov's inequality,

$$\underset{S}{\Pr}\left( \left| \|Sx\|_2^2 - 1 \right| > \epsilon \right) < \frac{\underset{S}{\mathbb{E}} \left| \|Sx\|_2^2 - 1 \right|^p}{\epsilon^p} < \delta.$$

The lemmas below show that Gaussian embeddings can be used to construct embeddings with the JL moment property.

**Lemma A.1** (Strong JL moment of Gaussian embeddings [18]). *Gaussian embeddings with $m = \Omega(\log(1/\delta)/\epsilon^2)$ satisfy the $(\epsilon, \delta)$-strong JL moment property.*

Below we review the composition rules of JL moment properties introduced in [1], which are used to prove the $(\epsilon, \delta)$-accurate sufficient condition in Theorem 3.1.

**Lemma A.2** (JL moment with Kronecker product). *If a matrix $S$ has the $(\epsilon, \delta, p)$-JL moment property, then the matrix $M = I_i \otimes S \otimes I_j$ also has the $(\epsilon, \delta, p)$-JL moment property for identity matrices $I_i$ and $I_j$ with any size. This relation also holds for the strong $(\epsilon, \delta)$-JL moment property.*

**Lemma A.3** (Strong JL moment with matrix product). *There exists a universal constant $L$, such that for any constants $\epsilon, \delta \in [0, 1]$ and any integer $k$, if $M_1 \in \mathbb{R}^{d_2 \times d_1}, \cdots, M_k \in \mathbb{R}^{d_{k+1} \times d_k}$ are independent random matrices, each having the strong $\left( \frac{\epsilon}{L\sqrt{k}}, \delta \right)$-JL moment property, then the product matrix $M = M_k \cdots M_1$ satisfies the strong $(\epsilon, \delta)$-JL moment property.*

## B  Definitions and basic properties of tensor network embedding

In this section, we introduce definitions and basic properties of tensor network embeddings. These properties will be used in Appendix C and Appendix D for detailed computational cost analysis. The notation defined in the main text is summarized in Table 4, which is also used in later analysis.

### B.1  Graph notation for tensor network and tensor contraction

We use undirected hypergraphs to represent tensor networks. For a given hypergraph $G = (V, E, w)$, $V$ represents the vertex set, $E$ represents the set of hyperedges, and $w$ is a function such that $w(e)$ is the natural logarithm of the tensor dimension size represented by the hyperedge $e \in E$. We use $E(u, v)$ to denote the set of hyperedges adjacent to both $u$ and $v$, which includes the edge $(u, v)$ and hyperedges adjacent to $u, v$. We use $E(A, B)$ to denote the set of hyperedges connecting two subsets $A, B$ of $V$ with $A \cap B = \emptyset$. We use $E(A, *)$ to denote all uncontracted edges only adjacent to $A$, $E(A, *) = \{(u) \in E : u \in A\}$. we illustrate $E(A, B), E(A, *)$ in Fig. 8. For any set $A \subseteq V$, we let

$$E(A) = E(A, V \setminus A) \cup E(A, *). \tag{B.1}$$

| Notations | Meanings |
| --- | --- |
| $S, S_i$ | Embedding matrix |
| $m$ | Sketch size |
| $G_E = (V_E, E_E, w)$ | Embedding tensor network |
| $G_D = (V_D, E_D, w)$ | Input data tensor network |
| $\bar{E} = \{e_1, \ldots, e_N\}$ | Set of edges to be sketched |
| $s_i$ | Size of $e_i$ in $\bar{E}$ |
| $T_0$ | Given data contraction tree |
| $\mathcal{D}(e_1), \ldots, \mathcal{D}(e_N), \mathcal{S}, \mathcal{I}$ | Subsets of contractions in $T_0$ |
| $X(e_i)$ | Sub network contracted by $\mathcal{D}(e_i)$ |

Table 4: Notations used throughout the paper.

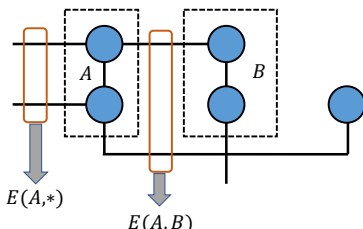

$E(A, *)$

$E(A, B)$

Figure 8: An example of $E(A, *)$ and $E(A, B)$, where both $A, B$ are subset of vertices.

A tensor network implicitly represents a tensor with a set of (small) tensors and a specific contraction pattern. We use $G[A] = (A, E_A, w)$ to denote a sub tensor network defined on $A \subseteq V$, where $E_A$ contains all hyperedges in $E$ adjacent to any $v \in A$.

Our analysis also use directed graphs to represent tensor network linearizations. We use $E(u, v)$ to denote the edge from $u$ to $v$, and similarly use $E(A, B)$ to denote the set of edges from $A$ to $B$.

When representing the contraction tree, we use $(v_1, v_2)$ to denote the contraction of $v_1, v_2$. This notation is also used to represent multiple contractions. For example, we use $(((v_1, v_4), (v_2, v_5)), v_3)$ to represent the contraction tree shown in Fig. 9. The computational cost of a contraction tree is the summation of each contraction's cost. In the discussion throughout the paper, we assume that all tensors in the network are dense. Therefore, the contraction of two general dense tensors $\mathcal{A}$ and $\mathcal{B}$, represented as vertices $v_a$ and $v_b$ in $G = (V, E, w)$, can be cast as a matrix multiplication, and the overall asymptotic cost is

$$\Theta(\exp(w(E(v_a)) + w(E(v_b)) - w(E(v_a, v_b))))$$

with classical matrix multiplication algorithms. In general, contracting tensor networks with arbitrary structure is #P-hard [11, 48].

Here is an example of constrained contraction tree, which is defined in Definition 1. Consider a tensor network with three tensors, $v_1, v_2, v_3$, with a given contraction tree $T_0$ that is $((v_1, v_2), v_3)$, which indicates that $v_1$ first contracts with $v_2$ and subsequently with $v_3$. Consider an additional tensor network consisting of $v_1, v_2, v_3$ and another tensors $u$. Then the contraction tree $(((v_1, v_2), u), v_3)$, $(((v_1, u), v_2), v_3)$ and $(((v_1, v_2), v_3), u)$ are all constrained on $T_0$, since the contraction ordering of $v_1, v_2, v_3$ remains unchanged. However, the contraction tree $(((v_1, v_3), u), v_2)$ is not constrained on $T_0$.

For a given data $G_D$ and its given contraction tree $T_0$, its dimension tree is a directed binary tree showing the way edges in $\bar{E}$ are merged onto the same tensor. Each vertex in the dimension tree is a subset $E' \subseteq \bar{E}$, and for any two vertices $E'_1, E'_2$ of the dimension tree with the same parent, there is

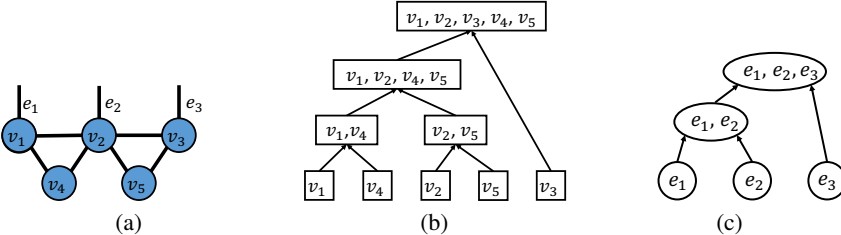

Figure 9: Example of a tensor network, its contraction tree and the corresponding dimension tree.

a contraction in $T_0$ such that the two input tensors are incident to $E_1', E_2'$, respectively. One example is shown in Fig. 9.

## B.2 Definitions used in the analysis of tensor network embedding

In this section, we introduce definitions that will be used in later analysis. For a (hyper)graph $G = (V, E, w)$ and two subsets of $V$ denoted as $A, B$, we define $\text{cut}_G(A, B) = \sum_{e \in E(A,B)} w(e)$. Similarly, we define $\text{cut}_G(A, *) = \sum_{e \in E(A,*)} w(e)$, and define $\text{cut}_G(A) = \sum_{e \in E(A)} w(e)$, where $E(A)$ is expressed in (B.1). When $G$ is a directed hypergraph, $\text{cut}_G(A, B)$ denotes the sum of the weights of edges from $A$ to $B$. When $G$ is an undirected graph, $\text{cut}_G(A, B)$ denotes the sum of the weights of hyperedges connecting $A$ and $B$.

For two tensors represented by two subsets $A, B \subset V$ and $A \cap B = \emptyset$, the logarithm of the contraction cost between a tensor represented by $A$ and a tensor represented by $B$, $(A, B)$, is

$$\text{cost}_G(A, B) = \text{cut}_G(A) + \text{cut}_G(B) - \text{cut}_G(A, B).$$

Note that the function cost is only defined on undirected hypergraphs.

Consider a given input data $G_D = (V_D, E_D, w)$ and an embedding $G_E = (V_E, E_E, w)$. Below we let $V = V_E \cup V_D$, $E = E_E \cup E_D$, and $G = (V, E, w)$ denote the hypergraph including both the embedding and the input data. We use $L = (V, E_E, w)$ to denote the graph including $V$ and all edges in the embedding, and use $R = (V, E \setminus E_E, w)$. Note that in this work we focus on the case where $L$ is a graph, and $R$ can be a general hypergraph. We illustrate $G, G_D, G_E, L, R$ in Fig. 10. For any $A, B \subset V$ and $A \cap B = \emptyset$, we have

$$\text{cut}_G(A) = \text{cut}_L(A) + \text{cut}_R(A), \tag{B.2}$$

and

$$\text{cut}_G(A, B) = \text{cut}_L(A, B) + \text{cut}_R(A, B). \tag{B.3}$$

Based on (B.2) and (B.3), we have

$$\text{cost}_G(A, B) = \text{cost}_L(A, B) + \text{cost}_R(A, B).$$

Our analysis of tensor network embedding is based on the linearization of the tensor network graph. Linearization casts an undirected graph into a *directed acyclic graph* (DAG). We define linearization formally below, then specify linearizations of the data and embedding graphs that our analysis considers.

**Definition 6** (Linearization DAG). A linearization of the undirected graph $G = (V, E, w)$ is defined by the DAG $G' = (V, E', w)$ induced by a given choice of vertex ordering in $V$. For each contracted edge in $E$, $E'$ contains an same-weight edge directing towards the higher indexed vertex. For each uncontracted edge in $E$, $E'$ contains an edge with the same weight that is directed outward from the vertex it is adjacent to.

Based on Definition 6, we define the *sketching linearization DAG*, $G_S = (V, E_S, w)$, as a DAG defined on top of the graph $L = (V, E_E, w)$, which includes all vertices in both the embedding and the data and all embedding edges. For a given vertex ordering of embedding vertices, $G_S$ is the linearization of $L$ based on the ordering with all data vertices being ordered ahead of embedding vertices.

As discussed in Section 3, for a given sketching linearization, the sketching accuracy of each tensor $\mathcal{A}_i$ at $v_i$ is dependent on the row size of its matricization $A_i$, which is the weighted size of the edge set adjacent to $v_i$ containing all uncontracted edges and contracted edges also adjacent to $v_j$ with $j > i$, which is called *effective sketch dimension* of $v_i$ throughout the paper. Based on the definition, when $v \in V_E$, $\mathsf{cut}_{G_S}(v)$ equals the effective sketch dimension size of $v$. When $v \in V_D$, $\mathsf{cut}_{G_S}(v)$ represents the size of the sketch dimension adjacent to $v$. We look at embeddings $G_E$ not only satisfying the $(\epsilon, \delta)$-accurate sufficient condition in Theorem 3.1, but also only have one output sketch dimension ($|E_1| = 1$) with the output sketch size $m = \Theta(N_E \log(1/\delta)/\epsilon^2)$. For each one of these embeddings, there must exist a linearization $G_S$ such that for all $v \in V_E$, we have

$$\mathsf{cut}_{G_S}(v) = \Omega(\log(m)). \tag{B.4}$$

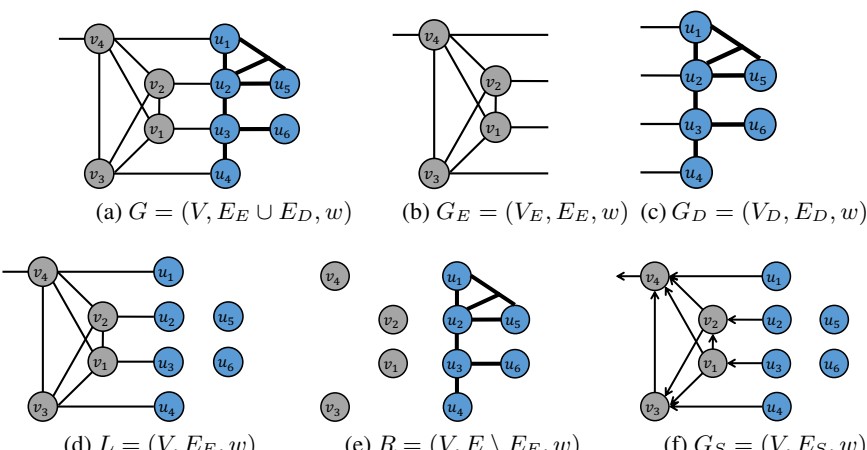

(a) $G = (V, E_E \cup E_D, w)$    (b) $G_E = (V_E, E_E, w)$   (c) $G_D = (V_D, E_D, w)$

(d) $L = (V, E_E, w)$      (e) $R = (V, E \setminus E_E, w)$      (f) $G_S = (V, E_S, w)$

Figure 10: Illustration of graphs and hypergraphs used throughout the paper.

## B.3 Properties of tensor network embedding

We now derive properties that are used in the sketching computational cost analysis. In Lemma B.1, we show relations between cuts in the graph $L$ and cuts in the graph $G_S$. In Lemma B.2, we show relations between costs in the graph $L$ and cuts in the graph $G_S$. Lemma B.2 along with cut lower bounds (B.4) is used to derive lower bounds for $\mathsf{cost}_L$ and $\mathsf{cost}_G$ in Appendix D.

**Lemma B.1.** *Consider an embedding $G_E = (V_E, E_E, w)$ and a data tensor network $G_D = (V_D, E_D, w)$, and a given sketching linearization $G_S = (V, E_S, w)$, where $V = V_E \cup V_D$. For any $A, B \subset V$ and $A \cap B = \emptyset$, the following relations hold,*

$$\mathsf{cut}_L(A) = \mathsf{cut}_{G_S}(A) + \mathsf{cut}_{G_S}(V \setminus A, A), \tag{B.5}$$

$$\mathsf{cut}_L(A, B) = \mathsf{cut}_{G_S}(A, B) + \mathsf{cut}_{G_S}(B, A), \tag{B.6}$$

$$\mathsf{cut}_{G_S}(A \cup B) = \mathsf{cut}_{G_S}(A) + \mathsf{cut}_{G_S}(B) - \mathsf{cut}_{G_S}(A, B) - \mathsf{cut}_{G_S}(B, A)$$
$$= \mathsf{cut}_{G_S}(A) + \mathsf{cut}_{G_S}(B) - \mathsf{cut}_L(A, B). \tag{B.7}$$

*Proof.* (B.5) and (B.6) hold directly based on the definition of the linearization DAG. For (B.7), based on (B.1), we have

$$\begin{aligned} \mathsf{cut}_{G_S}(A \cup B) &= \mathsf{cut}_{G_S}(A \cup B, V \setminus (A \cup B)) + \mathsf{cut}_{G_S}(A \cup B, *) \\ &= \mathsf{cut}_{G_S}(A, V \setminus (A \cup B)) + \mathsf{cut}_{G_S}(B, V \setminus (A \cup B)) + \mathsf{cut}_{G_S}(A \cup B, *) \\ &= \mathsf{cut}_{G_S}(A, V \setminus A) - \mathsf{cut}_{G_S}(A, B) + \mathsf{cut}_{G_S}(B, V \setminus B) - \mathsf{cut}_{G_S}(B, A) \\ &\quad + \mathsf{cut}_{G_S}(A, *) + \mathsf{cut}_{G_S}(B, *) \\ &= \mathsf{cut}_{G_S}(A) + \mathsf{cut}_{G_S}(B) - \mathsf{cut}_{G_S}(A, B) - \mathsf{cut}_{G_S}(B, A). \end{aligned} \tag{B.8}$$

Note that the second and third equalities in (B.8) hold since $A$ and $B$ are disjoint sets. This finishes the proof. □

**Lemma B.2.** *Consider any data $G_D = (V_D, E_D, w)$ and embedding $G_E = (V_E, E_E, w)$, and a sketching linearization $G_S = (V, E_S, w)$, where $V = V_D \cup V_E$. For any two subsets $A, B \in V$ such that $A \cap B = \emptyset$, the contraction of two tensors that are the contraction outputs of $A$ and $B$ has a logarithm cost of*

$$\text{cost}_L(A, B) = \text{cut}_{G_S}(A) + \text{cut}_{G_S}(B) + \text{cut}_{G_S}(V \setminus (A \cup B), A \cup B).$$

*Proof.* Based on Lemma B.1, we have

$$\text{cut}_L(A) \overset{\text{(B.5)}}{=} \text{cut}_{G_S}(V \setminus A, A) + \text{cut}_{G_S}(A), \tag{B.9}$$

$$\text{cut}_L(B) \overset{\text{(B.5)}}{=} \text{cut}_{G_S}(V \setminus B, B) + \text{cut}_{G_S}(B). \tag{B.10}$$

Based on (B.6), we have

$$\begin{aligned}
&\text{cut}_{G_S}(V \setminus A, A) + \text{cut}_{G_S}(V \setminus B, B) - \text{cut}_L(A, B) \\
&= \text{cut}_{G_S}(V \setminus A, A) + \text{cut}_{G_S}(V \setminus B, B) - \text{cut}_{G_S}(A, B) - \text{cut}_{G_S}(B, A) \\
&= \text{cut}_{G_S}(V \setminus (A \cup B), A) + \text{cut}_{G_S}(V \setminus (A \cup B), B) \\
&= \text{cut}_{G_S}(V \setminus (A \cup B), A \cup B). \tag{B.11}
\end{aligned}$$

Based on (B.9),(B.10), (B.11), we have

$$\begin{aligned}
\text{cost}_L(A, B) &= \text{cut}_L(A) + \text{cut}_L(B) - \text{cut}_L(A, B) \\
&= \text{cut}_{G_S}(A) + \text{cut}_{G_S}(B) + \text{cut}_{G_S}(V \setminus (A \cup B), A \cup B).
\end{aligned}$$

This finishes the proof. □

**Lemma B.3.** *Consider any data $G_D = (V_D, E_D, w)$ and an embedding $G_E = (V_E, E_E, w)$, and a sketching linearization $G_S = (V, E_S, w)$ such that the embedding is $(\epsilon, \delta)$-accurate. Then for any $U \subseteq V$ such that there exists $v \in U$ and $\text{cut}_{G_S}(v) \geq \log(m)$, we have $\text{cut}_{G_S}(U) \geq \log(m)$.*

*Proof.* When $U$ is a subset of the data vertices, $U \subseteq V_D$, this holds directly since

$$\text{cut}_{G_S}(U) = \sum_{u \in U} \text{cut}_{G_S}(u) \geq \text{cut}_{G_S}(v) \geq \log(m).$$

Next we consider the case where $U \cap V_E \neq \emptyset$. Let $A = U \cap V_E$ and $B = U \cap V_D$. Based on the definition of DAG, there is no directed cycle in the subgraph $G_S[A]$. Therefore, there exists one vertex $s \in A$, such that $\text{cut}_{G_S}(s, A \setminus \{s\}) = 0$. Based on Lemma B.1, we have

$$\begin{aligned}
\text{cut}_{G_S}(A) \overset{\text{(B.7)}}{=} &\ \text{cut}_{G_S}(s) + \text{cut}_{G_S}(A \setminus \{s\}) - \text{cut}_{G_S}(s, A \setminus \{s\}) - \text{cut}_{G_S}(A \setminus \{s\}, s) \\
\geq &\ \text{cut}_{G_S}(s) - \text{cut}_{G_S}(s, A \setminus \{s\}) \\
= &\ \text{cut}_{G_S}(s) \overset{\text{(B.4)}}{\geq} \log(m),
\end{aligned}$$

In addition, we have $\text{cut}_{G_S}(A, B) = 0$ since $A \subseteq V_E$ and $B \subseteq V_D$. Thus we have

$$\begin{aligned}
\text{cut}_{G_S}(U) = \text{cut}_{G_S}(A \cup B) \overset{\text{(B.7)}}{=} &\ \text{cut}_{G_S}(A) + \text{cut}_{G_S}(B) - \text{cut}_{G_S}(A, B) - \text{cut}_{G_S}(B, A) \\
= &\ \text{cut}_{G_S}(A) + \text{cut}_{G_S}(B) - \text{cut}_{G_S}(B, A) \\
\geq &\ \text{cut}_{G_S}(A) \geq \log(m).
\end{aligned}$$

This finishes the proof. □

## C  Computationally-efficient sketching algorithm

In this section, we introduce the detail of the computationally-efficient sketching algorithm in Algorithm 1. Consider a given data tensor network $G_D = (V_D, E_D, w)$ and a given data contraction tree, $T_0$. Also let $N_D = |V_D|$, and let $\bar{E} \subseteq E_D$ denote the set of edges to be sketched, and $N = |\bar{E}|$. Below we let $\bar{E} = \{e_1, e_2, \ldots, e_N\}$, and let each $e_i$ has weight $\log(s_i) > \log(m)$. Based on the

definition we have $N \leq N_D$. Let one contraction path representing $T_0$ be expressed as a sequence of $N_D - 1$ contractions,

$$\{(U_1, V_1), \ldots, (U_{N_D-1}, V_{N_D-1})\}. \tag{C.1}$$

Above we use $(U_i, V_i)$ to represent the contraction of two intermediate tensors represented by two subset of vertices $U_i, V_i \subset V_D$. Below we let

$$\begin{aligned}
a_i &= \exp(\mathsf{cut}_R(U_i) - \mathsf{cut}_R(U_i, V_i)), \\
c_i &= \exp(\mathsf{cut}_R(V_i) - \mathsf{cut}_R(U_i, V_i)), \\
d_i &= \exp(\mathsf{cut}_R(U_i \cup V_i))/(a_i c_i), \\
b_i &= \exp(\mathsf{cut}_R(U_i, V_i))/d_i. \tag{C.2}
\end{aligned}$$

Note that $d_i$ represents the size of uncontracted dimensions adjacent to both $U_i$ and $V_i$, and $b_i$ represents the size of contracted dimensions between $U_i$ and $V_i$. We also have $\mathsf{cost}_R(U_i, V_i) = \log(a_i b_i c_i d_i)$, and $\mathsf{cut}_R(U_i \cup V_i) = \log(a_i c_i d_i)$. $a_i, b_i, c_i, d_i$ are visualized in Fig. 11.

In Section 4, the $N_D - 1$ contractions are categorized into $N + 2$ sets, $\mathcal{D}(e_1), \ldots, \mathcal{D}(e_N), \mathcal{S}, \mathcal{I}$, where $\mathcal{D}(e_i)$ contains contractions such that $e_i$ is the only data edge adjacent to the contraction output and in $\bar{E}$, $\mathcal{S}$ contains contractions $(U_i, V_i)$ such that both $U_i$ and $V_i$ are adjacent to edges in $\bar{E}$, and $\mathcal{I}$ includes $(U_i, V_i)$ such that both $U_i$ and $V_i$ are not adjacent to $\bar{E}$. An illustration of these sets is provided below.

Consider an input data consisting of five tensors, $v_1, v_2, v_3, v_4, v_5$, where $v_1$ is adjacent to the edge $e_1$, $v_2$ is adjacent to $e_2$, $v_3$ is adjacent to $e_3$, and $e_1, e_2, e_3$ are the edges to be sketched. There are no edges to be sketched adjacent to $v_4, v_5$. Consider the contraction tree $(((v_1, v_4), (v_2, v_5)), v_3)$, where $v_1$ contracts with $v_4$ and outputs $v_{1,4}$, $v_2$ contracts with $v_5$ and outputs $v_{2,5}$, and then $v_{1,4}, v_{2,5}$ contract together into $v_{1,2,4,5}$, and $v_{1,2,4,5}$ contracts with $v_3$. We have $\mathcal{D}(e_1) = \{(v_1, v_4)\}$ and $\mathcal{D}(e_2) = \{(v_2, v_5)\}$, since $v_1$, $v_2$ are adjacent to $e_1, e_2$, respectively. We also have $\mathcal{I} = \emptyset$ and $\mathcal{D}(e_3) = \emptyset$, since each contraction is adjacent to at least one edge in $\{e_1, e_2, e_3\}$, and there is no contraction such that $e_3$ is the only data edge in the output. All the remaining contractions are in $\mathcal{S}$, so $\mathcal{S} = \{(v_{1,4}, v_{2,5}), (v_{1,2,4,5}, v_3)\}$.

### C.1 Sketching with the embedding containing a binary tree of small tensor networks

We now present the details of applying the embedding containing a binary tree of small tensor networks. In Section 4, we define $\mathcal{S}$ as the set containing contractions $(U_i, V_i)$ such that both $U_i$ and $V_i$ are adjacent to edges in $\bar{E}$. For each contraction $i \in \mathcal{S}$, one small embedding tensor network (denoted as $Z_i$) is applied to the contraction. Let $\hat{U}_i, \hat{V}_i$ denote the sketched $U_i$ and $V_i$ formed in previous contractions in the sketching contraction tree $T_B$, such that $\hat{U}_i \cap V_D = U_i$ and $\hat{V}_i \cap V_D = V_i$. The structure of $Z_i$ is determined so that the asymptotic cost to sketch $(\hat{U}_i, \hat{V}_i)$ is minimized, under the constraint that $Z_i$ is $(\epsilon/\sqrt{N}, \delta)$-accurate and only has one output sketch dimension.

The structure of $Z_i$ is illustrated in Fig. 11. For the case $a_i \leq c_i$, the structure is shown in Fig. 11a, and sketching is performed via the contraction sequence of contracting $\hat{U}_i$ and $v_1$ first, then with $\hat{V}_i$, and then with $v_2$ (also denoted as a contraction sequence of $(((\hat{U}_i, v_1), \hat{V}_i), v_2)$). For the case $a_i > c_i$, the structure of $Z_i$ is shown in Fig. 11b, and the sketching is performed via the contraction sequence of $(((\hat{V}_i, v_2), \hat{U}_i), v_1)$. With this algorithm, sketching $(\hat{U}_i, \hat{V}_i)$ yields a computational cost proportional to

$$y_i = a_i b_i c_i d_i m^2 + m^2 d_i \sqrt{a_i b_i c_i m} \cdot \min(\sqrt{a_i}, \sqrt{c_i}). \tag{C.3}$$

We show in Lemma D.6 that the asymptotic cost lower bound to sketch $(U_i, V_i)$ is also $\Omega(y_i)$.

### C.2 Computational cost analysis

We provide the computational cost analysis of Algorithm 1 in this section.

**Theorem C.1.** *Algorithm 1 has an asymptotic computational cost of*

$$\Theta\left(\sum_{j=1}^{N} t(e_j) + \sum_{i \in \mathcal{S}} y_i + \sum_{i \in \mathcal{I}} z_i\right), \tag{C.4}$$

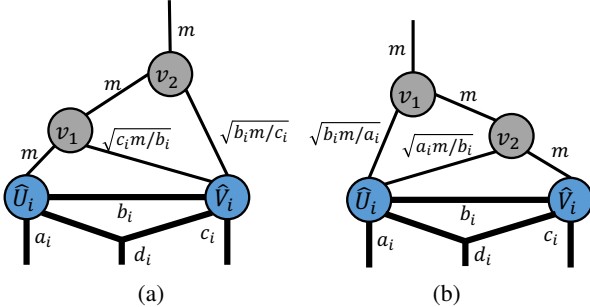

(a)            (b)

Figure 11: Illustration of the small network in the binary tree structured embedding. For each edge $e$, we show the dimension size of that edge (exponential in $w(e)$).

where $t(e_j)$ is the optimal asymptotic cost to sketch the sub tensor network $X(e_j)$ (defined in Table 4) with a matrix in the Kronecker product embedding, $y_i$ is expressed in (C.3), and

$$z_i = a_i b_i c_i d_i \cdot \min(\exp(\mathsf{cut}_{G_S}(U_i \cup V_i)), m), \tag{C.5}$$

where $a_i, b_i, c_i, d_i$ are expressed in (C.2).

*Proof.* The terms $\sum_{j \in N} t(e_j) + \sum_{i \in S} y_i$ can be easily verified based on the analysis in Section 4 and Appendix C.1.

Consider the contractions in $\mathcal{I}$, which include $(U_i, V_i)$ such that both $U_i$ and $V_i$ are not adjacent to $\bar{E}$, and contractions where $U_i$ or $V_i$ is adjacent to at least two edges in $\bar{E}$. The first type of contractions in $\mathcal{I}$ would have a cost of $\Theta(a_i b_i c_i d_i)$, and not be affected by previous sketching steps. For the second type, application of the Kronecker product and binary tree embeddings to $U_i$ and $V_i$ would reduce all adjacent edges in $\bar{E}$ to a single dimension of size $m$. Consequently, the contraction cost would be $\Theta(a_i b_i c_i d_i \cdot m)$. Summarizing both cases prove the cost in (C.5). The cost in (C.4) follows from combining the terms $\sum_{j \in N} t(e_j) + \sum_{i \in S} y_i$ and $\sum_{i \in \mathcal{I}} z_i$. $\qquad \square$

For the special case where each vertex in the data tensor network is adjacent to an edge to be sketched, we have $\mathcal{D}(e_j) = \emptyset$ for all $j \in [N]$ and $\mathcal{I} = \emptyset$, thus all the contractions are in the set $\mathcal{S}$. Therefore, sketching each $e_j$ has an asymptotic cost of $\Theta(t(e_j)) = \Theta(\exp(\mathsf{cut}_G(v_j)) \cdot m)$, where $v_j$ is the vertex in the data graph adjacent to $e_j$, and Theorem C.1 implies that the sketching cost would be

$$\Theta \left( \sum_{j=1}^{N} t(e_j) + \sum_{i \in S} y_i + \sum_{i \in \mathcal{I}} z_i \right) = \Theta \left( \sum_{j=1}^{N} \exp(\mathsf{cut}_G(v_j)) \cdot m + \sum_{j=1}^{N-1} y_i \right). \tag{C.6}$$

As we will show in Theorem D.1, this cost matches the asymptotic cost lower bound, when the embedding satisfies the $(\epsilon, \delta)$-accurate sufficient condition and only has one output sketch dimension.

When the data has a Kronecker product structure, we have $\mathsf{cut}_G(v_j) = w(e_j) = \log(s_j)$, and $a_i, b_i, c_i, d_i = 1$ for all $i \in \{1, \ldots, N-1\}$ for all contraction trees. Therefore,

$$y_i = a_i b_i c_i d_i m^2 + m^2 d_i \sqrt{a_i b_i c_i m} \cdot \min(\sqrt{a_i}, \sqrt{c_i}) = m^2 + m^{2.5},$$

and the sketching cost is

$$\Theta \left( \sum_{j=1}^{N} s_j m + N m^{2.5} \right). \tag{C.7}$$

As we will show in Appendix E, sketching with tree tensor network embeddings yield an asymptotic cost of $\Theta \left( \sum_{j=1}^{N} s_j m + N m^3 \right)$. Therefore, Algorithm 1 is more efficient to sketch Kronecker product input data.

# D   Lower bound analysis

In this section, we discuss the *asymptotic* computational lower bound for sketching with embeddings satisfying the $(\epsilon, \delta)$-accurate sufficient condition and only have one output sketch edge. In Appendix D.1, we discuss the case where the data has uniform sketch dimensions. In this case, each vertex in the data tensor network is adjacent to an edge to be sketched. In Appendix D.2, we discuss the sketching computational lower bound for a more general case, when the data tensor network can have arbitrary graph structure, and vertices not adjacent to sketch edges are allowed. For both cases, we assume that the size of each dimension to be sketched is greater than the sketch size.

## D.1   Sketching data with uniform sketch dimensions

We now discuss the sketching asymptotic cost lower bound when the data $G_D = (V_D, E_D, w)$ has uniform sketch dimensions, where each $v \in V_D$ is adjacent to an edge to be sketched with size lower bounded by the target sketch size, $m$. We have $N = |\bar{E}| = |V_D|$, and we let the size of each $e_i \in \bar{E}$ be denoted $s_i > m$. We let $V = V_E \cup V_D$ denote the set of all vertices in both the data and the embedding. Below, we show the main theorem using lemmas and notations introduced in Appendix B.

**Theorem D.1.** *For any embedding $G_E$ satisfying the $(\epsilon, \delta)$-accurate sufficient condition and only has one output sketch dimension, and any contraction tree $T_B$ of $(G_D, G_E)$ constrained on the data contraction tree $T_0$ expressed in (4.2), the sketching asymptotic cost is lower bounded by*

$$\Omega\left(\sum_{j=1}^{N} \exp(\text{cut}_G(v_j)) \cdot m + \sum_{j=1}^{N-1} y_i\right), \tag{D.1}$$

*where $m = \Omega(N \log(1/\delta)/\epsilon^2)$ represents the embedding sketch size, $v_j$ is the vertex in $V_D$ adjacent to $e_j$, and $\exp(\text{cut}_G(v_j))$ denotes the size of the tensor at $v_j$, and $y_i$ is expressed in (C.3).*

We present the proof of Theorem D.1 at the end of Appendix D.1. Note that the first term in (D.1), $\sum_{v \in V_D} \exp(\text{cut}_G(v)) \cdot m$, is a term independent of the data contraction tree, while the second term is dependent of the data contraction tree.

*Proof of Theorem 4.1.* The asymptotic cost of of Algorithm 1 in (C.6) matches the lower bound shown in Theorem D.1, thus proving the statement. ☐

Theorem D.1 also yields an asymptotic lower bound for sketching data with a Kronecker product structure. We state the results below.

**Corollary D.2.** *Consider an input data $G_D$ representing a vector with a Kronecker product structure and each $v_j$ for $j \in [N]$ is adjacent to an edge to be sketched with size $s_j$. For any embedding $G_E$ satisfying the $(\epsilon, \delta)$-accurate sufficient condition with only one output sketch dimension and any contraction tree $T_B$ of $(G_D, G_E)$, the asymptotic cost must be lower bounded by*

$$\Omega\left(\sum_{j=1}^{N} s_j m + N m^{2.5}\right),$$

*where $m = \Omega(N \log(1/\delta)/\epsilon^2)$.*

Below, we present some lemmas needed to prove Theorem D.1.

**Lemma D.3.** *Consider an $(\epsilon, \delta)$-accurate embedding $G_E = (V_E, E_E, w)$ with a sketching linearization $G_S = (V, E_S, w)$. Then for any subset of the embedding and data graph vertex set, $W \subseteq V$, we have $\text{cut}_{G_S}(W) \geq \log(m)$.*

*Proof.* Since each vertex in the data graph is adjacent to an edge to be sketched, and the edge dimension size is greater than $m$, we have $\text{cut}_{G_S}(w) \geq \log(m)$ for all $w \in V_D$. Since the embedding satisfies the $(\epsilon, \delta)$-accurate sufficient condition, we have $\text{cut}_{G_S}(w) \geq \log(m)$ for all $w \in V_E$. Therefore, $\text{cut}_{G_S}(w) \geq \log(m)$ for all $w \in V$. Based on Lemma B.3, $\text{cut}_{G_S}(W) \geq \log(m)$ for all $W \subseteq V$. ☐

**Lemma D.4.** *Consider an $(\epsilon, \delta)$-accurate embedding $G_E$ with a sketching linearization $G_S = (V, E_S, w)$. Consider any contraction tree $T_B$ for $(G_D, G_E)$. If there exists a contraction output of $U \subset V$ formed in $T_B$ and $\mathsf{cut}_{G_S}(U) > \log(m)$, then the asymptotic cost for the contraction tree $T_B$ must be lower bounded by $\Omega(\exp(\mathsf{cut}_R(U) + \mathsf{cut}_{G_S}(U)) \cdot m)$.*

*Proof.* Since $\mathsf{cut}_{G_S}(U) > \log(m)$, there must exist a contraction $(U, W) \in T_B$ with $W$ containing some vertices in $V \setminus U$. Based on Lemma D.3, $\mathsf{cut}_{G_S}(W) \geq \log(m)$. Based on Lemma B.2, we have

$$
\begin{aligned}
\mathsf{cost}_G(U, W) &= \mathsf{cost}_R(U, W) + \mathsf{cost}_L(U, W) \\
&= \mathsf{cost}_R(U, W) + \mathsf{cut}_{G_S}(U) + \mathsf{cut}_{G_S}(W) + \mathsf{cut}_{G_S}(V \setminus (U \cup W), U \cup W).
\end{aligned}
$$

Further, since $\mathsf{cost}_R(U, W) \geq \mathsf{cut}_R(U)$,

$$
\begin{aligned}
\mathsf{cost}_G(U, W) &\geq \mathsf{cut}_R(U) + \mathsf{cut}_{G_S}(U) + \mathsf{cut}_{G_S}(W) \\
&\geq \mathsf{cut}_R(U) + \mathsf{cut}_{G_S}(U) + \log(m).
\end{aligned}
$$

This proves the lemma since the contraction cost is $\Theta(\exp(\mathsf{cost}_G(U, W)))$. $\square$

In Lemma D.6, we show that when the data contraction tree $T_0$ contains the contraction $(U_i, V_i)$, then any contraction tree $T_B$ of $(G_D, G_E)$ that is constrained on $T_0$ will yield a contraction cost of $\Omega(y_i)$. To show that, we first discuss the case where $T_B$ also contains the contraction $(U_i, V_i)$ in Lemma D.5. The more general case where the contraction $(U_i, V_i)$ need not be in $T_B$ is discussed in Lemma D.6.

**Lemma D.5.** *Consider a specific contraction tree $T_0$ for $G_D$, where the contraction $(U_i, V_i)$ is in $T_0$. For any embedding $G_E$ satisfying the $(\epsilon, \delta)$-accurate sufficient condition with only one output sketch dimension and any contraction tree $T_B$ of $(G_D, G_E)$ constrained on $T_0$, if $(U_i, V_i)$ is also in $T_B$, the sketching asymptotic cost must be lower bounded by $\Omega(a_i b_i c_i d_i m^2 + a_i c_i d_i m^3)$, where $a_i, b_i, c_i, d_i$ are defined in (C.2).*

*Proof.* Consider any sketching linearization $G_S = (V, E_S, w)$ such that the embedding satisfies the $(\epsilon, \delta)$-accurate sufficient condition with only one output sketch dimension. Based on Lemma D.3, we have $\mathsf{cut}_{G_S}(U_i), \mathsf{cut}_{G_S}(V_i) \geq \log(m)$. Based on Lemma B.2, we have

$$
\begin{aligned}
\mathsf{cost}_G(U_i, V_i) &= \mathsf{cost}_R(U_i, V_i) + \mathsf{cost}_L(U_i, V_i) \\
&= \mathsf{cost}_R(U_i, V_i) + \mathsf{cut}_{G_S}(U_i) + \mathsf{cut}_{G_S}(V_i) + \mathsf{cut}_{G_S}(V \setminus (U_i \cup V_i), U_i \cup V_i) \\
&\geq \mathsf{cost}_R(U_i, V_i) + \mathsf{cut}_{G_S}(V_i) + \mathsf{cut}_{G_S}(V_i) \\
&\geq \log(a_i b_i c_i d_i) + 2\log(m).
\end{aligned}
$$

Thus this contraction has a cost of $\Omega(a_i b_i c_i d_i \cdot m^2)$. In addition, since $U_i, V_i$ are subsets of the data vertices, $\mathsf{cut}_L(U_i, V_i) = 0$. Therefore, based on (B.7),

$$
\mathsf{cut}_{G_S}(U_i \cup V_i) = \mathsf{cut}_{G_S}(U_i) + \mathsf{cut}_{G_S}(V_i) \geq 2\log(m).
$$

Based on Lemma D.4, the cost needed to sketch $U_i \cup V_i$ is $\Omega(a_i c_i d_i m^3)$. Thus the overall asymptotic cost is lower bounded by $\Omega(a_i b_i c_i d_i m^2 + a_i c_i d_i m^3)$. This finishes the proof. $\square$

**Lemma D.6.** *Consider a specific contraction tree $T_0$ for $G_D$, where the contraction $(U_i, V_i)$ is in $T_0$. For any embedding $G_E$ satisfying the $(\epsilon, \delta)$-accurate sufficient condition with only one output sketch dimension and any contraction tree $T_B$ of $(G_D, G_E)$ constrained on $T_0$, the sketching asymptotic cost must be lower bounded by*

$$
\Omega(y_i) = \Omega\left(a_i b_i c_i d_i m^2 + m^2 d_i \sqrt{a_i b_i c_i m} \cdot \min(\sqrt{a_i}, \sqrt{c_i})\right), \tag{D.2}
$$

*where $a_i, b_i, c_i, d_i$ are defined in (C.2), and $y_i$ is defined in (C.3).*

*Proof.* Consider any sketching linearization $G_S = (V, E_S, w)$ such that the embedding satisfies the $(\epsilon, \delta)$-accurate sufficient condition with only one output sketch dimension. We first consider the

case where the contraction $(U_i, V_i)$ exists in $T_B$. Based on Lemma D.5, the overall asymptotic cost is lower bounded by $\Omega\big(a_i b_i c_i d_i m^2 + a_i c_i d_i m^3\big)$. Since

$$a_i b_i c_i d_i m^2 + a_i c_i d_i m^3 = a_i c_i d_i m^2 (b_i + m) \geq 2 a_i c_i d_i m^2 \sqrt{b_i m},$$

the overall asymptotic cost is lower bounded by

$$\Omega\Big(a_i b_i c_i d_i m^2 + a_i c_i d_i m^2 \sqrt{b_i m}\Big) = \Omega\Big(a_i b_i c_i d_i m^2 + a_i m^2 d_i \sqrt{b_i c_i m}\Big),$$

and hence it satisfies (D.2).

We next consider the other case where the contraction $(U_i, V_i)$ is not performed directly in $T_B$. Since $T_B$ is constrained on $T_0$, there must exist a contraction $(\hat{U}_i, \hat{V}_i) \in T_B$ with either $\hat{U}_i$ or $\hat{V}_i$ containing embedding vertices, and $\hat{U}_i \cap V_D = U_i$, $\hat{V}_i \cap V_D = V_i$. Let $x$ be the last embedding vertex (based on the linearization order) applied in $T_B$ to $\hat{U}_i \cup \hat{V}_i$, so that $\mathrm{cut}_{G_S}(x, (\hat{U}_i \cup \hat{V}_i) \setminus \{x\}) = 0$. For the case where $x \in \hat{U}_i \setminus U_i$, we show below that the sketching asymptotic cost is lower bounded by

$$\Omega\Big(a_i b_i c_i d_i m^2 + a_i m^2 d_i \sqrt{b_i c_i m}\Big). \tag{D.3}$$

For the other case where $x \in \hat{V}_i \setminus V_i$, we have the cost is lower bounded by $\Omega\big(a_i b_i c_i d_i m^2 + c_i m^2 d_i \sqrt{a_i b_i m}\big)$ by symmetry. Together, these two results prove the lemma.

**Detailed proof of** (D.3)   Since $|\hat{U}_i| > |U_i|$, there must exist a contraction $(Y_1, Y_2)$, for which the output is $\hat{U}_i = Y_1 \cup Y_2$. Based on Lemma B.2, we have

$$\begin{aligned}
\mathrm{cost}_G(Y_1, Y_2) &= \mathrm{cost}_R(Y_1, Y_2) + \mathrm{cost}_L(Y_1, Y_2) \\
&= \mathrm{cost}_R(Y_1, Y_2) + \mathrm{cut}_{G_S}(Y_1) + \mathrm{cut}_{G_S}(Y_2) + \mathrm{cut}_{G_S}(V \setminus (Y_1 \cup Y_2), Y_1 \cup Y_2) \\
&\geq \mathrm{cost}_R(Y_1, Y_2) + \mathrm{cut}_{G_S}(Y_1) + \mathrm{cut}_{G_S}(Y_2) + \mathrm{cut}_{G_S}(\hat{V}_i, \hat{U}_i) \\
&\geq \log(a_i b_i d_i) + 2\log(m) + \mathrm{cut}_{G_S}(\hat{V}_i, \hat{U}_i).
\end{aligned}$$

Thus, the cost of the contraction $(Y_1, Y_2)$ is lower bounded by

$$\Omega\Big(a_i b_i d_i m^2 \cdot \exp\Big(\mathrm{cut}_{G_S}(\hat{V}_i, \hat{U}_i)\Big)\Big). \tag{D.4}$$

In addition, since

$$\begin{aligned}
\mathrm{cost}_G(\hat{U}_i, \hat{V}_i) &= \mathrm{cost}_R(\hat{U}_i, \hat{V}_i) + \mathrm{cut}_{G_S}(\hat{U}_i) + \mathrm{cut}_{G_S}(\hat{V}_i) + \mathrm{cut}_{G_S}(V \setminus (\hat{U}_i \cup \hat{V}_i), \hat{U}_i \cup \hat{V}_i) \\
&\geq \log(a_i b_i c_i d_i) + 2\log(m),
\end{aligned}$$

the contraction $(\hat{U}_i, \hat{V}_i)$ yields a cost lower bounded by

$$\Omega\big(a_i b_i c_i d_i \cdot m^2\big). \tag{D.5}$$

Combining (D.4) and (D.5), we have that the contractions $(Y_1, Y_2)$ and $(\hat{U}_i, \hat{V}_i)$ have a cost of

$$\Omega\Big(a_i b_i d_i m^2 \cdot \exp\Big(\mathrm{cut}_{G_S}(\hat{V}_i, \hat{U}_i)\Big) + a_i b_i c_i d_i \cdot m^2\Big). \tag{D.6}$$

When $\mathrm{cut}_{G_S}(\hat{V}_i, \hat{U}_i) = \log(m)$, (D.6) implies that the overall asymptotic cost is lower bounded by $\Omega(a_i b_i c_i d_i m^2 + a_i b_i d_i m^3)$. Since

$$a_i b_i c_i d_i m^2 + a_i b_i d_i m^3 = a_i b_i d_i m^2 (c_i + m) \geq 2 a_i b_i d_i m^2 \sqrt{c_i m},$$

the overall asymptotic cost is lower bounded by

$$\Omega\big(a_i b_i c_i d_i m^2 + a_i b_i d_i m^2 \sqrt{c_i m}\big) = \Omega\Big(a_i b_i c_i d_i m^2 + a_i m^2 d_i \sqrt{b_i c_i m}\Big).$$

When $\mathrm{cut}_{G_S}(\hat{V}_i, \hat{U}_i) < \log(m)$, based on Lemma B.1, the effective sketch dimensions of $\hat{U}_i \cup \hat{V}_i$ satisfy

$$\begin{aligned}
\mathrm{cut}_{G_S}(\hat{U}_i \cup \hat{V}_i) &= \Big(\mathrm{cut}_{G_S}(\hat{U}_i) - \mathrm{cut}_{G_S}(\hat{U}_i, \hat{V}_i)\Big) + \mathrm{cut}_{G_S}(\hat{V}_i) - \mathrm{cut}_{G_S}(\hat{V}_i, \hat{U}_i) \\
&\geq \mathrm{cut}_{G_S}(x) + \mathrm{cut}_{G_S}(\hat{V}_i) - \mathrm{cut}_{G_S}(\hat{V}_i, \hat{U}_i) \\
&\geq 2\log(m) - \mathrm{cut}_{G_S}(\hat{V}_i, \hat{U}_i),
\end{aligned} \tag{D.7}$$

where the first inequality holds since

$$\mathsf{cut}_{G_S}(\hat{U}_i) - \mathsf{cut}_{G_S}(\hat{U}_i, \hat{V}_i) = \mathsf{cut}_{G_S}(\hat{U}_i, V \setminus (\hat{U}_i \cup \hat{V}_i)) + \mathsf{cut}_{G_S}(\hat{U}_i, *)$$
$$\geq \mathsf{cut}_{G_S}(x, V \setminus (\hat{U}_i \cup \hat{V}_i)) + \mathsf{cut}_{G_S}(x, *) = \mathsf{cut}_{G_S}(x),$$

and the second inequality in (D.7) holds since $\mathsf{cut}_{G_S}(x), \mathsf{cut}_{G_S}(\hat{V}_i) \geq \log(m)$ based on Lemma D.3. Based on the condition $\mathsf{cut}_{G_S}(\hat{V}_i, \hat{U}_i) < \log(m)$ as well as (D.7), we have $\mathsf{cut}_{G_S}(\hat{U}_i \cup \hat{V}_i) > \log(m)$.

Based on Lemma D.4, since $\mathsf{cut}_{G_S}(\hat{U}_i \cup \hat{V}_i) > \log(m)$, there must exist another contraction in $T_B$ to sketch $\hat{U}_i \cup \hat{V}_i$ with a cost of

$$\Omega\Big(\exp\Big(\mathsf{cut}_R(\hat{V}_i \cup \hat{U}_i) + \mathsf{cut}_{G_S}(\hat{V}_i \cup \hat{U}_i)\Big) \cdot m\Big) = \Omega\Big(a_i c_i d_i \cdot \exp\Big(\mathsf{cut}_{G_S}(\hat{V}_i \cup \hat{U}_i)\Big) \cdot m\Big)$$
$$\overset{\text{(D.7)}}{=} \Omega\Bigg(a_i c_i d_i \cdot \frac{m^3}{\exp(\mathsf{cut}_{G_S}(\hat{V}_i, \hat{U}_i))}\Bigg).$$

Let $\alpha = \exp\Big(\mathsf{cut}_{G_S}(\hat{V}_i, \hat{U}_i)\Big)$, the asymptotic cost is then lower bounded by

$$\Omega\Big(a_i b_i c_i d_i m^2 + a_i b_i d_i m^2 \cdot \alpha + a_i c_i d_i m^3 \cdot \frac{1}{\alpha}\Big) = \Omega\Big(a_i b_i c_i d_i m^2 + a_i m^2 d_i \sqrt{b_i c_i m}\Big).$$

This finishes the proof.

$\square$

*Proof of Theorem D.1.* Based on Lemma D.6, the cost of $\Omega(y_i)$ is needed to sketch the contraction $(U_i, V_i)$. Since $T_0$ contains contractions $(U_i, V_i)$ for $i \in [N-1]$, the asymptotic cost of $T_B$ must be lower bounded by $\Omega\Big(\sum_{i=1}^{N-1} y_i\Big)$. In addition, in the analysis of Lemma D.6, at least one embedding vertex is needed to sketch each contraction $(U_i, V_i)$, thus $N_E = \Omega(N)$ and $m = \Omega(N \log(1/\delta)/\epsilon^2)$ for the lower bound $\Omega\Big(\sum_{i=1}^{N-1} y_i\Big)$ to hold.

In addition, each $v_j \in V_D$ for $j \in [N]$ is adjacent to $e_j$ and each $w(e_j) > \log(m)$. Based on Lemma D.4, the asymptotic cost must be lower bounded by

$$\Omega\left(\sum_{j=1}^{N} \exp(\mathsf{cut}_R(v_j) + \mathsf{cut}_{G_S}(v_j)) \cdot m\right) = \Omega\left(\sum_{j=1}^{N} \exp(\mathsf{cut}_G(v_j)) \cdot m\right).$$

The above holds since $v_j$ is a vertex in the data graph, thus $\mathsf{cut}_{G_S}(v_j) = \mathsf{cut}_L(v_j)$ and $\mathsf{cut}_R(v_j) + \mathsf{cut}_{G_S}(v_j) = \mathsf{cut}_G(v_j)$. This finishes the proof. $\square$

## D.2 Sketching general data

In this section, we look at general tensor network data $G_D$, where each vertex in $G_D$ can either be adjacent to an edge to be sketched with weight greater than $\log(m)$ or not adjacent to any edge to be sketched. Below we consider any data contraction tree $T_0$ containing $\mathcal{D}(e_1), \ldots, \mathcal{D}(e_N), \mathcal{S}, \mathcal{I}$ defined in Section 4. We also let $X(e_j) \subset V$ represent the sub network contracted by $\mathcal{D}(e_j)$. We present the asymptotic sketching lower bound in Theorem D.9.

**Lemma D.7.** *Consider $G_D$ with a data contraction tree $T_0$ containing $\mathcal{D}(e_j)$, which is a set containing contractions such that $e_j$ is the only data edge adjacent to the contraction output and in $\bar{E}$ (set of data edges to be sketched). For any embedding $G_E$ satisfying the $(\epsilon, \delta)$-accurate sufficient condition with only one output sketch dimension and any contraction tree $T_B$ of $(G_D, G_E)$ constrained on the data contraction tree $T_0$, the sketching asymptotic cost must be lower bounded by $\Omega(t(e_j))$, where $t(e_j)$ is the optimal asymptotic cost to sketch the sub tensor network $X(e_j)$ (defined in Table 4) with an adjacent matrix in the Kronecker product embedding.*

*Proof.* When $\mathcal{D}(e_j) = \emptyset$, $X(e_j) = \{v_j\}$, where $v_j$ is the vertex in the data graph adjacent to $e_j$. As is analyzed in the proof of Theorem D.1, the asymptotic cost must be lower bounded by

$\Omega(\exp(\mathsf{cut}_G(v_j)) \cdot m)$, which equals the asymptotic cost to contract $v_j$ with the adjacent embedding matrix.

Now we discuss the case where $\mathcal{D}(e_j) \neq \emptyset$. We first consider the case where there is a contraction $(X(e_j), W)$ in $T_B$. We show that under this case, the cost is lower bounded by $\Omega(t(e_j))$. We then show that for the case where there is no contraction $(X(e_j), W)$ in $T_B$, meaning that some sub network of $X(e_j)$ is sketched, the cost is also lower bounded by $\Omega(t(e_j))$. Summarizing both cases prove the lemma.

Consider the case where there exists a contraction $(X(e_j), W)$ in $T_B$. Contracting $X(e_j)$ yields a cost of $\Omega(\sum_{i \in \mathcal{D}(e_j)} a_i b_i c_i d_i \cdot s_j)$. Next we analyze the contraction cost of $(X(e_j), W)$. Since $X(e_j)$ is the contraction output of $\mathcal{D}(e_j)$, $W$ must either contain embedding vertices, or contain some data vertex adjacent to edges in $\bar{E}$ (edges to be sketched). Therefore, $W$ contains some vertex $v$ with $\mathsf{cut}_{G_S}(v) \geq \log(m)$. Based on Lemma B.3, we have $\mathsf{cut}_{G_S}(W) \geq \log(m)$. Therefore,

$$\mathsf{cost}_L(X(e_j), W) = \mathsf{cut}_{G_S}(X(e_j)) + \mathsf{cut}_{G_S}(W) + \mathsf{cut}_{G_S}(V \setminus (X(e_j) \cup W), X(e_j) \cup W)$$
$$\geq \log(s_j) + \log(m).$$

Let $l \in \mathcal{D}(e_j)$ denote the last contraction in $\mathcal{D}(e_j)$, then we have $\mathsf{cut}_R(X(e_j)) = \log(a_l c_l d_l)$. Thus, we have

$$\mathsf{cost}_G(X(e_j), W) = \mathsf{cost}_L(X(e_j), W) + \mathsf{cost}_R(X(e_j), W) \tag{D.8}$$
$$\geq \mathsf{cost}_L(X(e_j), W) + \mathsf{cut}_R(X(e_j)) \geq \log(a_l c_l d_l s_j m), \tag{D.9}$$

making the cost lower bounded by $\Omega\left(\sum_{i \in \mathcal{D}(e_j)} a_i b_i c_i d_i \cdot s_j + a_l c_l d_l s_j m\right)$. Note that contracting $X(e_j)$ and sketching the contraction output with a matrix in the Kronecker product embedding yields a cost of $\Theta\left(\sum_{i \in \mathcal{D}(e_j)} a_i b_i c_i d_i \cdot s_j + a_l c_l d_l s_j m\right)$, which upper-bounds the value of $t(e_j)$ based on definition. Thus the sketching cost is lower bounded by $\Omega(t(e_j))$.

Below we analyze the case where there is no contraction $(X(e_j), W)$ in $T_B$. Without loss of generality, for each contraction $(U_i, V_i)$ with $i \in \mathcal{D}(e_j)$, assume that $U_i$ is adjacent to $e_j$. When $X(e_j)$ is not formed in $T_B$, there must exist $U_k$ with $k \in \mathcal{D}(e_j)$, and a contraction $(U_k, X)$ with $X \subset V_E$ is in $T_B$. All contractions before $k$ yield a cost of

$$\Omega\left(\sum_{i \in \mathcal{D}(e_j), i < k} a_i b_i c_i d_i \cdot s_j\right). \tag{D.10}$$

Similar to the analysis for the contraction $(X(e_j), W)$ in (D.8), the contraction $(U_k, X)$ yields a cost of

$$\Omega(a_k b_k d_k s_j m). \tag{D.11}$$

For other contractions in $T_0$, $(U_i, V_i)$ with $i \in \mathcal{D}(e_j), i \geq k$, there must exist some contractions $(\hat{U}_i, \hat{V}_i)$ in $T_B$ with $\hat{U}_i \cap V_D = U_i$, $\hat{V}_i \cap V_D = V_i$, since $T_B$ is constrained on $T_0$. Therefore, we have

$$\mathsf{cost}_G(\hat{U}_i, \hat{V}_i) = \mathsf{cost}_R(\hat{U}_i, \hat{V}_i) + \mathsf{cost}_L(\hat{U}_i, \hat{V}_i) = \mathsf{cost}_R(U_i, V_i) + \mathsf{cost}_L(\hat{U}_i, \hat{V}_i)$$
$$\geq \mathsf{cost}_R(U_i, V_i) + \mathsf{cut}_{G_S}(\hat{U}_i) \geq \log(a_i b_i c_i d_i m). \tag{D.12}$$

In the last inequality in (D.12) we use the fact that there exists a vertex $v \in U_i \subseteq \hat{U}_i$ with $\mathsf{cut}_{G_S}(v) = \log(s_j) \geq \log(m)$, then based on Lemma B.3, $\mathsf{cut}_{G_S}(\hat{U}_i) \geq \log(m)$.

Combining (D.10), (D.11) and (D.12), we have the cost is lower bounded by

$$\Omega(f(k)) = \Omega\left(\sum_{i \in \mathcal{D}(e_j), i < k} a_i b_i c_i d_i \cdot s_j + a_k b_k d_k s_j m + \sum_{i \in \mathcal{D}(e_j), i \geq k} a_i b_i c_i d_i \cdot m\right),$$

where $f(k)$ represents the asymptotic cost to contract $X(e_j)$ with an embedding matrix adjacent to $e_j$, when sketching is performed at $k$th contraction with $k \in \mathcal{D}(e_j)$. Based on the definition of $t(e_j)$, we have $f(k) = \Omega(t(e_j))$, thus finishing the proof. $\qquad \square$

**Lemma D.8.** *Consider any data $G_D$. For any embedding $G_E$ satisfying the $(\epsilon, \delta)$-accurate sufficient condition with only one output sketch dimension and any contraction tree $T_B$ of $(G_D, G_E)$, the sketching asymptotic cost must be lower bounded by $\Omega(Nm^{2.5})$, where $m = \Omega(N\log(1/\delta)/\epsilon^2)$.*

*Proof.* Let $G_D'$ be the data with the same set of sketching edges $(\bar{E})$ as $G_D$, but $G_D'$ is a Kronecker product data. For any given contraction tree $T_B$ of $(G_D, G_E)$, there must exist a contraction tree of $(G_D', G_E)$ whose asymptotic cost is upper bounded by the cost of $T_B$. Therefore, the asymptotic cost lower bound to contract $(G_D', G_E)$ must also be the asymptotic cost lower bound to contract $(G_D, G_E)$. Based on Corollary D.2, the asymptotic cost of $T_B$ must be lower bounded by

$$\Omega\left(\sum_{j=1}^{N} s_j m + Nm^{2.5}\right) = \Omega(Nm^{2.5}).$$

$\square$

**Theorem D.9.** *For any embedding $G_E$ satisfying the $(\epsilon, \delta)$-accurate sufficient condition and any contraction tree $T_B$ of $(G_D, G_E)$ constrained on the data contraction tree $T_0$ expressed in (4.2), the sketching asymptotic cost must be lower bounded by*

$$\Omega\left(\sum_{j=1}^{N} t(e_j) + \sum_{i \in \mathcal{S}} a_i b_i c_i d_i m^2 + Nm^{2.5} + \sum_{i \in \mathcal{I}} z_i\right), \tag{D.13}$$

*where $m = \Omega(N\log(1/\delta)/\epsilon^2)$, $a_i, b_i, c_i, d_i$ are expressed in (C.2), $t(e_j)$ is the optimal asymptotic cost to sketch the sub tensor network $X(e_j)$ (the sub network contracted by $\mathcal{D}(e_j)$, also defined in Table 4) with an adjacent matrix in the Kronecker product embedding, and $z_i$ is expressed in (C.5).*

*Proof.* The term $\sum_{j=1}^{N} t(e_j)$ can be proven based on Lemma D.7, and the term $Nm^{2.5}$ with $m = \Omega(N\log(1/\delta)/\epsilon^2)$ can be proven based on Lemma D.8. Below we show the asymptotic cost is also lower bounded by $\Omega(\sum_{i \in \mathcal{S}} a_i b_i c_i d_i m^2 + \sum_{i \in \mathcal{I}} z_i)$, thus finishing the proof.

For each contraction $(U_i, V_i)$ in $T_0$ with $i \in \mathcal{S} \cup \mathcal{I}$, there must exist a contraction $(\hat{U}_i, \hat{V}_i)$ in $T_B$, and $\hat{U}_i \cap V_D = U_i$, $\hat{V}_i \cap V_D = V_i$. For the case where $i \in \mathcal{S}$, since both $U_i$ and $V_i$ contain edges to be sketched, we have $\mathsf{cut}_{G_S}(\hat{U}_i) \geq \log(m)$ and $\mathsf{cut}_{G_S}(\hat{V}_i) \geq \log(m)$ based on Lemma B.3. Therefore, we have

$$\sum_{i \in \mathcal{S}} \mathsf{cost}_G(\hat{U}_i, \hat{V}_i) = \sum_{i \in \mathcal{S}} \mathsf{cost}_R(\hat{U}_i, \hat{V}_i) + \mathsf{cost}_L(\hat{U}_i, \hat{V}_i)$$

$$= \sum_{i \in \mathcal{S}} \mathsf{cost}_R(U_i, V_i) + \mathsf{cost}_L(\hat{U}_i, \hat{V}_i)$$

$$\geq \sum_{i \in \mathcal{S}} \mathsf{cost}_R(U_i, V_i) + \mathsf{cut}_{G_S}(\hat{U}_i) + \mathsf{cut}_{G_S}(\hat{V}_i)$$

$$\geq \sum_{i \in \mathcal{S}} \log(a_i b_i c_i d_i m^2), \tag{D.14}$$

where the first inequality above holds based on Lemma B.2. This shows the cost is lower bounded by $\Omega(\sum_{i \in \mathcal{S}} a_i b_i c_i d_i m^2)$.

Now consider the case where $i \in \mathcal{I}$. In this case, either $\mathsf{cut}_{G_S}(U_i \cup V_i) = 0$ or $\mathsf{cut}_{G_S}(U_i \cup V_i) \geq \log(m)$. When $\mathsf{cut}_{G_S}(U_i \cup V_i) = 0$, we have $\mathsf{cut}_{G_S}(\hat{U}_i \cup \hat{V}_i) \geq \mathsf{cut}_{G_S}(U_i \cup V_i)$. When $\mathsf{cut}_{G_S}(U_i \cup V_i) \geq \log(m)$, based on Lemma B.3, we have $\mathsf{cut}_{G_S}(\hat{U}_i \cup \hat{V}_i) \geq \log(m)$. To summarize, we have

$$\mathsf{cut}_{G_S}(\hat{U}_i \cup \hat{V}_i) \geq \min(\mathsf{cut}_{G_S}(U_i \cup V_i), \log(m)),$$

thus

$$\sum_{i \in \mathcal{I}} \mathsf{cost}_G(\hat{U}_i, \hat{V}_i) = \sum_{i \in \mathcal{I}} \mathsf{cost}_R(\hat{U}_i, \hat{V}_i) + \mathsf{cost}_L(\hat{U}_i, \hat{V}_i)$$

$$\geq \sum_{i \in \mathcal{I}} \mathsf{cost}_R(U_i, V_i) + \mathsf{cut}_{G_S}(\hat{U}_i) + \mathsf{cut}_{G_S}(\hat{V}_i)$$

$$\geq \sum_{i \in \mathcal{I}} \mathsf{cost}_R(U_i, V_i) + \mathsf{cut}_{G_S}(\hat{U}_i \cup \hat{V}_i)$$

$$\geq \sum_{i \in \mathcal{I}} \log(a_i b_i c_i d_i) + \min(\mathsf{cut}_{G_S}(U_i \cup V_i), \log(m))$$

$$= \sum_{i \in \mathcal{I}} \log(z_i). \tag{D.15}$$

This shows the sketching cost is lower bounded by $\Omega\big(\sum_{i \in \mathcal{I}} z_i\big)$, thus finishing the proof. $\qquad\square$

*Proof of Theorem 4.2.* Based on Theorem C.1, the computational cost of Algorithm 1 is

$$\alpha = \Theta\left(\sum_{j=1}^{N} t(e_j) + \sum_{i \in \mathcal{S}} y_i + \sum_{i \in \mathcal{I}} z_i\right).$$

Let $\beta$ equals the expression in (D.13). We have

$$\frac{\alpha}{\beta} = \frac{\Theta\big(\sum_{j=1}^{N} t(e_j) + \sum_{i \in \mathcal{S}} y_i + \sum_{i \in \mathcal{I}} z_i\big)}{\Omega\big(\sum_{j=1}^{N} t(e_j) + \sum_{i \in \mathcal{S}}(a_i b_i c_i d_i m^2 + m^{2.5}) + \sum_{i \in \mathcal{I}} z_i\big)} = O\left(\frac{\sum_{i \in \mathcal{S}} y_i}{\sum_{i \in \mathcal{S}}(a_i b_i c_i d_i m^2 + m^{2.5})}\right)$$

$$= O\left(\max_{i \in \mathcal{S}} \frac{y_i}{a_i b_i c_i d_i m^2 + m^{2.5}}\right) = O\left(\max_{i \in \mathcal{S}} \frac{a_i b_i c_i d_i m^2 + m^2 d_i \sqrt{a_i b_i c_i m} \cdot \min(\sqrt{a_i}, \sqrt{c_i})}{a_i b_i c_i d_i m^2 + m^{2.5}}\right)$$

$$= O(1) + O\left(\max_{i \in \mathcal{S}} \frac{m^2 d_i \sqrt{a_i b_i c_i m} \cdot \min(\sqrt{a_i}, \sqrt{c_i})}{a_i b_i c_i d_i m^2 + m^{2.5}}\right).$$

Below we derive asymptotic upper bound of the term $\theta = \frac{m^2 d_i \sqrt{a_i b_i c_i m} \cdot \min(\sqrt{a_i}, \sqrt{c_i})}{a_i b_i c_i d_i m^2 + m^{2.5}}$. We analyze the case below with $a_i \leq c_i$, and the other case with $a_i > c_i$ can be analyzed in a similar way based on the symmetry of $a_i, c_i$ in $\theta$.

When $a_i \leq c_i$, we have $m^2 d_i \sqrt{a_i b_i c_i m} \cdot \min(\sqrt{a_i}, \sqrt{c_i}) = a_i m^2 d_i \sqrt{b_i c_i m}$. We consider two cases, one satisfies $\sqrt{b_i c_i m} \leq b_i c_i$ and the other satisfies $\sqrt{b_i c_i m} > b_i c_i$.

When $\sqrt{b_i c_i m} \leq b_i c_i$, we have $\theta \leq 1$, thus $\frac{\alpha}{\beta} = O(1)$, thus satisfying the theorem statement.

When $\sqrt{b_i c_i m} > b_i c_i$, which means that $m > b_i c_i$, we have

$$\theta = \frac{a_i m^2 d_i \sqrt{b_i c_i m}}{a_i b_i c_i d_i m^2 + m^{2.5}} \leq \min\left(\frac{\sqrt{m}}{\sqrt{b_i c_i}}, a_i d_i \sqrt{b_i c_i}\right) \leq \sqrt{m},$$

thus $\frac{\alpha}{\beta} \leq O(\sqrt{m})$. In addition, when $G_D$ is a graph, we have $d_i = 1$ for all $i$. Therefore,

$$\theta \leq \min\left(\frac{\sqrt{m}}{\sqrt{b_i c_i}}, a_i d_i \sqrt{b_i c_i}\right) = \min\left(\frac{\sqrt{m}}{\sqrt{b_i c_i}}, a_i \sqrt{b_i c_i}\right)$$

$$\leq \min\left(\frac{\sqrt{m}}{\sqrt{b_i c_i}}, c_i \sqrt{b_i c_i}\right) \leq \min\left(\frac{\sqrt{m}}{(b_i c_i)^{1/2}}, (b_i c_i)^{3/2}\right) \leq m^{0.375}.$$

Therefore, in this case we have $\frac{\alpha}{\beta} \leq O(m^{0.375})$, thus finishing the proof. $\qquad\square$

# E   Analysis of tree tensor network embeddings

In this section, we provide detailed analysis of sketching with tree embeddings. The algorithm to sketch with tree embedding is similar to Algorithm 1, and the only difference is that for each

contraction $(U_i, V_i)$ with $i \in \mathcal{S}$, such that both $U_i$ and $V_i$ are adjacent to edges in $\bar{E}$, we sketch it with one embedding tensor $z_i$ rather than a small network. Let $\hat{U}_i, \hat{V}_i$ denote the sketched $U_i$ and $V_i$ formed in previous contractions in the sketching contraction tree $T_B$, such that $\hat{U}_i \cap V_D = U_i$ and $\hat{V}_i \cap V_D = V_i$, we sketch $(\hat{U}_i, \hat{V}_i)$ via the contraction path $((\hat{U}_i, \hat{V}_i), z_i)$. For the case where each vertex in the data tensor network is adjacent to an edge to be sketched, the sketching cost would be

$$\Theta\left( \sum_{j=1}^{N} \exp(\mathsf{cut}_G(v_j)) \cdot m + \sum_{j=1}^{N-1} (a_i b_i c_i d_i m^2 + a_i c_i d_i m^3) \right), \tag{E.1}$$

where $v_j$ is the vertex in the data graph adjacent to $e_j$, $a_i, b_i, c_i, d_i$ are defined in (C.2), and we replace the term $y_i = a_i b_i c_i d_i m^2 + m^2 d_i \sqrt{a_i b_i c_i m} \cdot \min(\sqrt{a_i}, \sqrt{c_i})$ in (C.6) with $a_i b_i c_i d_i m^2 + a_i c_i d_i m^3$.

*Proof of Theorem 4.3.* Since each contraction in $T_0$ contracts dimensions with size being at least the sketch size $m$, we have $b_i \geq m$ for $i \in \mathcal{S}$. Therefore,

$$m^2 d_i \sqrt{a_i b_i c_i m} \cdot \min(\sqrt{a_i}, \sqrt{c_i}) \leq a_i m^2 d_i \sqrt{b_i c_i m} \leq a_i b_i d_i m^2 \sqrt{c_i} \leq a_i b_i c_i d_i m^2,$$

and the asymptotic cost in (C.6) would be

$$\Theta\left( \sum_{j=1}^{N} \exp(\mathsf{cut}_G(v_j)) \cdot m + \sum_{j=1}^{N-1} a_i b_i c_i d_i m^2 \right). \tag{E.2}$$

Based on Theorem D.1, (E.2) matches the sketching asymptotic cost lower bound for this data. Since $a_i c_i d_i m^3 \leq a_i b_i c_i d_i m^2$ so (E.1) equals (E.2), sketching with tree embeddings also yield the optimal asymptotic cost. $\square$

When the data has a Kronecker product structure, sketching with tree tensor network embedding is less efficient compared to Algorithm 1. As is shown in (E.3), Algorithm 1 yields a cost of $\Theta\left(\sum_{j=1}^{N} s_j m + N m^{2.5}\right)$ to sketch the Kronecker product data. However, for tree embeddings, the asymptotic cost (E.1) is equal to

$$\Theta\left( \sum_{j=1}^{N} s_j m + N m^3 \right). \tag{E.3}$$

# F  Computational cost analysis of sketched CP-ALS

In this section, we provide detailed computational cost analysis of the sketched CP-ALS algorithm based on Algorithm 1. We are given a tensor $\mathcal{X} \in \mathbb{R}^{s \times \cdots \times s}$, and aim to decompose that into $N$ factor matrices, $A_i \in \mathbb{R}^{s \times R}$ for $i \in [N]$. Let $L_i = A_1 \odot \cdots \odot A_{i-1} \odot A_{i+1} \odot \cdots \odot A_N$ and $R_i = X_{(i)}^T$. In each iteration, we aim to update $A_i$ via solving a sketched linear least squares problem, $A_i = \underset{A}{\operatorname{argmin}} \left\| S_i L_i A^T - S_i R_i \right\|_F^2$, where $S_i$ is an embedding constructed based on Algorithm 1.

Below we first discuss the sketch size of $S_i$ sufficient to make each sketched least squares problem accurate. We then discuss the contraction trees of $L_i$, on top of which embedding structures are determined. We select contraction trees such that contraction intermediates can be reused across subproblems. Finally, we present the detailed computational cost analysis of the sketched CP-ALS algorithm.

### F.1  Sketch size sufficient for accurate least squares subproblem

Since the tensor network of $L_i$ contains $N$ output dimensions and $L_i$ contains $R$ columns, we show below that a sketch size of $\Theta(NR \log(1/\delta)/\epsilon^2) = \tilde{\Theta}(NR/\epsilon^2)$ is sufficient for the least squares problem to be $(\epsilon, \delta)$-accurate.

**Theorem F.1.** *Consider the sketched linear least squares problem $\min_A \left\| S_i L_i A^T - S_i R_i \right\|_F^2$. Let $S_i$ be an embedding constructed based on Algorithm 1, with the sketch size $m = \Theta(NR \log(1/\delta)/\epsilon^2)$, solving the sketched least squares problem gives us an $(1+\epsilon)$-accurate solution with probability at least $1 - \delta$.*

*Proof.* Algorithm 1 outputs an embedding with $\Theta(N)$ vertices. Based on Theorem 3.1, a sketched size of $\Theta(NR \log(1/\delta)/\epsilon^2)$ will make the embedding $(\epsilon, \frac{\delta}{e^R})$-accurate. Based on the $\epsilon$-net argument [50], $S$ is the $(\epsilon, \delta)$-accurate subspace embedding for a subspace with dimension $R$. Therefore, we can get an $(1+\epsilon)$-accurate solution with probability at least $1 - \delta$ for the least squares problem. $\qquad\square$

### F.2 Data contraction trees and efficient embedding structures

The structures of embeddings $S_1, \ldots, S_N$ also depend on the data contraction trees for $L_1, \ldots, L_N$. We denote the contraction tree of $L_i$ as $T_i$. We construct $T_i$ for $i \in [N]$ such that resulting embeddings $S_1, \ldots, S_N$ have common parts, which yields more efficient sketching computational cost via reusing contraction intermediates.

Let the vertex $v_i$ represent the matrix $A_i$. We also let $V_L^{(i)} = \{v_1, \ldots, v_i\}$ denote the set of all first $i$ vertices, and let $V_R^{(i)} = \{v_i, \ldots, v_N\}$ denote the set of vertices from $v_i$ to $v_N$. In addition, we let $\mathcal{C}_L^{(i)}$ denote a contraction tree to fully contract $V_L^{(i)}$, from $v_1$ to $v_i$. Let $\mathcal{C}_L^{(1)} = \emptyset$, we have for all $i \geq 1$, $\mathcal{C}_L^{(i+1)} = \mathcal{C}_L^{(i)} \cup \left\{ (V_L^{(i)}, v_{i+1}) \right\}$. Similarly, we let $\mathcal{C}_R^{(i)}$ denote a contraction tree to fully contract $V_R^{(i)}$, from $v_N$ to $v_i$. Let $\mathcal{C}_R^{(N)} = \emptyset$, we have for all $i \leq N$, $\mathcal{C}_R^{(i-1)} = \mathcal{C}_R^{(i)} \cup \left\{ (V_R^{(i)}, v_{i-1}) \right\}$.

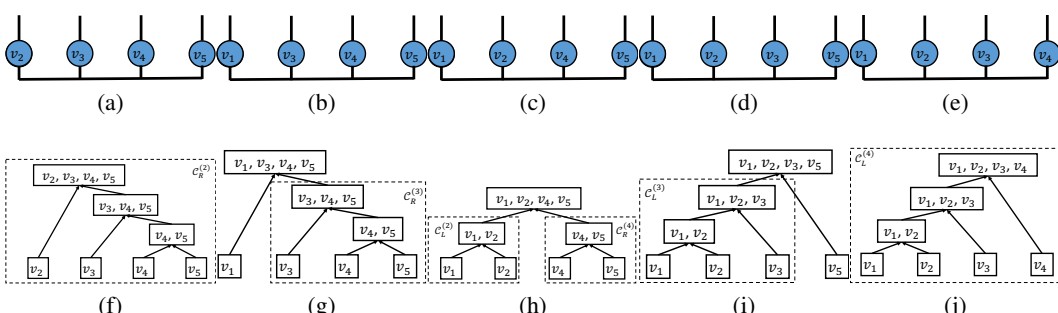

Figure 12: (a)-(e): Representations of $L_1, \ldots, L_5$ for the CP decomposition of an order 5 tensor. (f)-(j): Data dimension trees $T_1, \ldots, T_5$.

Note that the vertex set of the tensor network of $L_i$ is $V_L^{(i-1)} \cup V_R^{(i+1)}$. Each $T_i$ is constructed so that $V_L^{(i-1)}, V_R^{(i+1)}$ are first contracted via the contraction trees $\mathcal{C}_L^{(i-1)}, \mathcal{C}_R^{(i+1)}$, respectively, then a contraction of $(V_L^{(i-1)}, V_R^{(i+1)})$ is used to contract them into a single tensor. We illustrate $T_i$ for the CP decomposition of an order 5 tensor in Fig. 12.

These tree structures allow us to reuse contraction intermediates during sketching. On top of $T_1$, sketching $L_1$ using Algorithm 1 yields a cost of $\Theta(N(smR + m^{2.5}R))$, where the term $\Theta(NsmR)$ comes from sketching with the Kronecker product embedding, and the term $\Theta(Nm^{2.5}R)$ comes from sketching each data contraction in $\mathcal{C}_R^{(2)}$. Since $\mathcal{C}_R^{(2)} = \mathcal{C}_R^{(3)} \cup \left\{ (V_R^{(3)}, v_2) \right\}$, all contractions in $\mathcal{C}_R^{(3)}$ are sketched, and we obtain the sketching output of $V_R^{(3)}$, which is denoted as $\hat{V}_R^{(3)}$ below.

We use $\hat{V}_R^{(3)}$ formed during sketching $L_1$ to sketch $L_2$. Since $T_2$ contains contractions

$$ T_2 = \mathcal{C}_R^{(3)} \cup \mathcal{C}_L^{(1)} \cup \left\{ (V_L^{(1)}, V_R^{(3)}) \right\} = \mathcal{C}_R^{(3)} \cup \left\{ (v_1, V_R^{(3)}) \right\}, $$

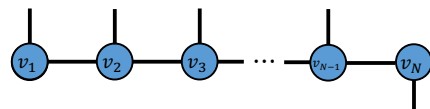

Figure 13: Illustration of the matricization of the tensor train ($X$) to be sketched. The $N-1$ uncontracted edges adjacent to $v_1, \ldots, v_{N-1}$ are to be sketched.

through reusing $\hat{V}_R^{(3)}$, we only need to sketch $(v_1, \hat{V}_R^{(3)})$ to compute $S_2 L_2$, which only costs $\Theta(smR + m^{2.5}R)$. Similarly, sketching each $L_i$ for $i \geq 2$ only costs $\Theta(smR + m^{2.5}R)$, thus making the overall cost of sketching $L_1, \ldots, L_N$ being $\Theta(N(smR + m^{2.5}R))$.

### F.3 Detailed algorithm and the overall computational cost

---

**Algorithm 2 Sketched-ALS**: Sketched ALS for CP decomposition

---

1: **Input:** Input tensor $\mathcal{X}$, initializations $A_1, \ldots, A_N$, maximum number of iterations $I_{\max}$
2: $G_D(L_i) \leftarrow$ structure of the data $L_i = A_1 \odot \cdots \odot A_{i-1} \odot A_{i+1} \odot \cdots \odot A_N$ for $i \in [N]$
3: $T_i \leftarrow$ contraction tree of $G_D(L_i)$ for $i \in [N]$ constructed based on Appendix F.2
4: Build tensor network embeddings $S_i$ on $G_D(L_i)$ and $T_i$ based on Algorithm 1 for $i \in [N]$
5: Compute $\hat{R}_i \leftarrow S_i X_{(i)}^T$ for $i \in [N]$
6: **for** $t \in [I_{\max}]$ **do**
7:    **for** $i \in [N]$ **do**
8:       Compute $\hat{L}_i \leftarrow S_i L_i$
9:       $A_i \leftarrow \underset{X}{\mathrm{argmin}} \left\| \hat{L}_i X - \hat{R}_i \right\|_F^2$
10:    **end for**
11: **end for**
12: **return** $A_1, \ldots, A_N$

---

We present the detailed sketched CP-ALS algorithm in Algorithm 2. Here we analyze the overall computational cost of the algorithm.

Line 5 yields a preparation cost of the algorithm. Note that we construct $S_i$ based on Appendix F.2, where they share common tensors. Contracting $S_1 X_{(1)}^T$ yields a cost of $\Theta(s^N m)$. On top of that, contracting $S_i X_{(i)}^T$ for $i \geq 2$ also only yields a cost of $\Theta(s^N m)$, making the overall preparation cost $\Theta(s^N m)$.

Within each ALS iteration (Lines 7-10), based on Appendix F.2, computing $S_i L_i$ for $i \in [N]$ costs $\Theta(N(smR + m^{2.5}R))$. For each $i \in [N]$, line 9 costs $\Theta(mR^2)$, making the cost of per-iteration least squares solves $\Theta(NmR^2)$. Based on Appendix F.1, a sketch size of $m = \tilde{\Theta}(NR/\epsilon^2)$ is sufficient for the least squares solution to be $(1 + \epsilon)$-accurate with probability at least $1 - \delta$. Overall, the per-iteration cost is $\Theta(N(smR + m^{2.5}R)) = \tilde{\Theta}(N^2(N^{1.5}R^{3.5}/\epsilon^3 + sR^2)/\epsilon^2)$.

## G   Computational cost analysis of sketching for tensor train rounding

We provide the computational cost lower bound analysis of computing $SX$, where $X$ denotes a matricization of the tensor train data shown in Fig. 13. This step is the computational bottleneck of the tensor train randomized rounding algorithm proposed in [10]. As is discussed in Section 5, we assume the tensor train has order $N$ with the output dimension sizes equal $s$, the tensor train rank is $R < s$, and the goal is to round the rank to $r < R$. The sketch size $m$ of $S$ is $r$ plus some constant, and is assumed to be smaller than $R$. The lower bound is derived within all embeddings satisfying the sufficient condition in Theorem 3.1 and only have one output sketch dimension with size $m$.

For the data contraction tree that contracts the tensor train shown in Fig. 13 from left to right, we have $a_i = 1, b_i = R, c_i = R, d_i = 1$ for $i \in [N-2]$, where $a_i, b_i, c_i, d_i$ are expressed in (C.2).

Based on Theorem D.9, the sketching asymptotic cost lower bound is

$$\Omega\left(\sum_{j=1}^{N-1} t(e_j) + \sum_{i \in \mathcal{S}} a_i b_i c_i d_i m^2 + Nm^{2.5} + \sum_{i \in \mathcal{I}} z_i\right) = \Omega\left(\sum_{j=1}^{N-1} t(e_j) + \sum_{i \in \mathcal{S}} a_i b_i c_i d_i m^2\right)$$

$$= \Omega\left(\sum_{j=1}^{N-1} \exp(\mathsf{cut}_G(v_j)) \cdot m + \sum_{j=1}^{N-2} a_i b_i c_i d_i m^2\right)$$

$$= \Omega\left(NsR^2 m + NR^2 m^2\right) = \Omega\left(NsR^2 m\right).$$

Above we use the fact that $\exp(\mathsf{cut}_G(v_1)) = sR$, and for $j \in \{2, \ldots, N-1\}$, we have $\exp(\mathsf{cut}_G(v_j)) = sR^2$. Sketching with Algorithm 1, tree embedding and tensor train embedding all would yield this optimal asymptotic cost.

## H  Additional experiments

| | Uniform | Gaussian |
|---|---|---|
| Tensor network embedding (Algorithm 1) | 85.0 | 78.1 |
| Tree embedding (Theorem 4.3) | 75.4 | 68.1 |
| Tensor train embedding [10] | 49.3 | 45.4 |

Table 5: Comparison of the mean sketch sizes with different input data distribution when sketching a tensor train input. The tensor train order is chosen to be 6 and the dimension size is chosen to be 500. Each reported sketch size is the mean of 25 experiments. Variables in the uniform distribution are within the interval of $[0, 1]$, and variables in the Gaussian distribution have the same mean as the uniform distribution and have the unit variance.

In this section, we experimentally verify that the sketch size of embeddings to get the same sketching accuracy trends similarly for both uniform and Gaussian input tensor distributions. We compare the performance of general tensor network embedding used in Algorithm 1, tree embedding discussed in Theorem 4.3, and the baseline, tensor train embedding [10], in sketching tensor train input data in Table 5. For each input tensor train $x$ and a specific embedding structure, we calculate the relative sketching error twice under different sketch sizes, and record the smallest sketch size such that both of its relative sketching errors are within 0.2, $\frac{\|Sx\|_2}{\|x\|_2} \leq 0.2$. As can be seen in the table, for both distributions, tensor network embedding produces a slightly larger sketch size than tree embedding, and tensor train embedding yields the lowest sketch size.