# OpenReview forum: "Cost-efficient Gaussian tensor network embeddings for tensor-structured inputs"
_NeurIPS.cc/2022/Conference — NeurIPS 2022 Accept_

### Official Review · Reviewer_bo7e · 2022-07-04

**Rating:** 6
**Confidence:** 4
**Soundness:** 3 good
**Presentation:** 2 fair
**Contribution:** 3 good

**Summary:**

This paper investigated an efficient sketching method for embedding the tensor with tensor network (TN) structures into lower-dimensional spaces. Compared with the existing works, the main contribution of this paper includes the points as follows:

- The authors established a systematic framework regarding how to sketch a TN with a lower-dimensional representation.
- The authors carefully discussed the computational cost of the proposed method with necessary proofs for demonstrating the efficiency.
- Two potential applications are introduced: a) ALS-CP algorithm; and b) tensor train rounding algorithm.

**Questions:**

1. As shown in Figure 4, the embedding network is a  binary tree-structured, of which the vertices are repeatedly represented by a small TN. How do you determine the structure (e.g., bond dimension, network topology) for these small TNs? What happens if another topology is applied as an alternative?
2. Could you give more interpretation, such as an understandable example, for Definition 1, the constrained contraction tree?
3. It would be appreciated if the authors can give more interpretation about the Defs for $D(e_1)$, $\mathcal{S,I}$ given in lines 241-250.
4. Regrading the computational cost, suppose the data TN itself is contractible, meaning that the contraction cost is polynomial to the tensor dimension, I am wondering if the whole sketching of the proposed method is contractible as well? In other words, is the contractibility preserved under sketching?
5. What is the difference between #P-complete and #P-hard? Are they equivalent?
6. In line 398, why are the entries of the tensor drawn from the uniform distribution, rather than Gaussian?
7. If possible, could you evaluate the sketching dimension (or compression ratio) numerically with varying the parameter $\epsilon$ in a smaller range? Although the authors mentioned that the setting of $\epsilon$ to be 0.1-0.2 might be good, I think it depends on the specific task.

**Limitations:**

The main limitation of this work is that the proposed method seems only to work superiorly when the data TN is very low-rank. It might not be suitable for most tasks in practice.

**Strengths And Weaknesses:**

**Strengths:**

- (originality) This work reveals the potential advantage regarding of how the inherent TN structure can be leveraged to improve the sketching performance.
- (originality) The joint discussion of the effective sketching with computational cost (modeled by the contraction tree) is very interesting.
- (quality) The theoretical discussion of the paper on complexity analysis is remarkable.

**Weaknesses:**

- (clarity) The paper is relatively well written, but the introduction of the algorithm details (sec. 4) is very hard to follow, even though I carefully go through the appendix. For example, the meaning of the letter U in Eq. (4.2) is not clear. I finally found its definition in the appendix but it should be well defined in the manuscript.
- (significance) A fundamental assumption of the work — the TN ranks should be smaller than the mode dimensions — seems not reasonable in practice. It might be true for some tensor decomposition models such as CP and Tucker for lower-order tensors. In the sense of TN, one typically has to be faced in practice with lower mode dimension (only 2 or 3) but higher ranks (e.g., 100). In this case (also acknowledged by the authors in the paper), the improvement by the proposed method becomes not so significant. The numerical results in the paper also verify this point.

---

> ### Author Response · Authors · 2022-08-02
> **Response to Reviewer bo7e**
>
> We would like to thank the reviewer for the constructive feedback and great questions! Our comments to your suggestions are as follows:
>
> Q: The intro of the alg details (sec. 4) is hard to follow, and the meaning of U in (4.2) is not clear:
>
> A: Thanks! $(U_i,V_i)$ in (4.2) represents the contraction of two intermediate tensors represented by two subsets of vertices $U_i,V_i\subset V_D$, as explained in line 760 of the appendix. We will move detailed explanations of the alg in appendix C to the main text in the revised version.
>
> Q: The assumption that the TN ranks should be smaller than the mode dimensions seems not reasonable in practice.
>
> A: Thanks! The paper does not assume that TN ranks are smaller than mode dimensions. Sec. 3 and 4 analyze general tensor network input data. One major contribution is that we proposed the sketching lower bound analysis and an algorithm to sketch so that the bound can be reached.
>
> Our alg is more efficient than existing ones for the low TN rank case, and comparable for the high-TN rank case where the existing algorithm already matches the cost lower bound and is efficient. Also, this is the first effort to offer a thorough lower-bound analysis, which helps build efficient sketching algorithms.
>
> Q: As to the structure of each small TN:
>
> A: Thanks! Due to page limits, we explain how to choose each small TN's structure in Appendix C.1. Each TN has 2 tensors to decrease the computational cost. If another topology is used, the asymptotic computational cost isn't optimal. If each TN only includes 1 tensor, the embedding is only computationally efficient for certain data structures, as shown in line 300, "Tree tensor network embedding efficiency."
>
> Q: Could you give more interpretation, such as an understandable example, for Definition 1, the constrained contraction tree?
>
> A: Thanks! Consider a tensor network with three tensors, $A,B,C$, and a given contraction tree $T_0$ that is $((A,B),C)$, meaning $A$ contracts with $B$ first, then with $C$. Consider another tensor network consisting of $A,B,C$ and another tensors $X$. Then the contraction tree $(((A,B),X),C)$, $(((A,X),B),C)$ and $(((A,B),C),X)$ all are constrained on $T_0$, since the contraction ordering of $A,B,C$ remain the same. However, the contraction tree $(((A,C),X), B)$ is not constrained on $T_0$.
>
> Q: It would be appreciated if the authors can give more interpretation about the Defs for ${D}(e_1),S,I$ given in lines 241-250.
>
> A: We'll include the example below in the revised version to help readers understand. Consider an input data with 5 tensors, $v_1,v_2,v_3,v_4,v_5$, where $v_1$ has a dimension $e_1,v_2$ has a dimension $e_2,v_3$ has a dimension $e_3$, and dimensions $e_1,e_2,e_3$ are to be sketched. $v_4,v_5$ have no dimension to be sketched. Consider the contraction tree $(((v_1,v_4),(v_2,v_5)),v_3)$, where $v_1$ contracts with $v_4$ and outputs $v_{1,4},v_2$ contracts with $v_5$ and outputs $v_{2,5}$, and then $v_{1,4},v_{2,5}$ contract together into $v_{1,2,4,5}$, and $v_{1,2,4,5}$ contracts with $v_3$. For this example, $D(e_1)$ has $(v_1,v_4),D(e_2)$ has $(v_2,v_5),D(e_3)=\emptyset$, and $I=\emptyset$. The remaining contractions are all in set $S$, which contain $(v_{1,4},v_{2,5})$ and $(v_{1,2,4,5},v_3).$
>
> Q: Suppose the data TN itself is contractible, is the whole sketching of the proposed method contractible as well?
>
> A: Yes, contractibility is preserved under sketching. For each contraction in the original network, in our algorithm, we either preserve this contraction or introduce a sketching matrix and contract at a lower cost. We will explain this in the revised version.
>
> Q: The difference between #P-complete and #P-hard.
>
> A: They are not equivalent, #P-hard problems may not be #P, and a problem is #P-complete when it's both #P and #P-hard  (as with any complexity class).
>
> Q: In line 398, why are the entries of the tensor drawn from the uniform distribution, rather than Gaussian?
>
> A: We observe uniform and Gaussian distributions with the same mean sketch similarly.  We will add this in the revised version.
> When sketching tensor train input data with order 6 and dimension size 500, to reach a relative error of 0.2, the sketch size mean across 250 experiments are as follows,
>
> |uniform|Gaussian
>
> TN embedding|85.0|78.1
>
> Tree embedding|75.4|68.1
>
> TT embedding|49.3|45.4
>
> Q: Could you evaluate the sketching dimension numerically by varying the parameter in a smaller range? Although the authors mentioned that the setting of $\epsilon$ to be 0.1-0.2 might be good, I think it depends on the specific task.
>
> A: Thanks! Below we show the sketch size-error relation with TN embedding. We can see the asymptotic scaling of sketch size is $O(1/\epsilon^2)$. Setting $\epsilon$ to be 0.1-0.2 would be good for CP-ALS, and it might not be suitable for all applications. We will explain this in the revised version.
>
> error|sketch size
>
> 0.8|11.54
>
> 0.4|30.65
>
> 0.2|85.265
>
> 0.18|99.165
>
> 0.14|124.51
>
> 0.1|194.09

---

### Official Review · Reviewer_QwJS · 2022-07-07

**Rating:** 5
**Confidence:** 2
**Soundness:** 3 good
**Presentation:** 2 fair
**Contribution:** 2 fair

**Summary:**

This paper discussed the tensor network embedding. Specially, they focus on the problem to derive a sequence of sketching matrices. They generalized the embedding accuracy in Theorem 3.1, w.r.t the sketch size of the matrix. They later proposed an efficient sketching algorithm which enjoyed batter per-iteration cost than previous methods.

**Questions:**

Though most questions are in weakness. I do curious that, Is there a good way to assume the hypergraph structure of the tensor network embedding?



**Strengths And Weaknesses:**

Strengths:
1. The theoretical analysis is strong. Especially theorem 3.1 which discussed the accuracy of the tensor network embedding.
2. The proposed sketching method has less computational cost per iteration.
3. This framework showed broad application on different tensor decompositions like CP decomposition and tensor train embedding.

Weaknesses:
1. The theoretical part of this paper discussed a very general tensor network embedding rooted in the hypergraph presentation. But in the later part, they barely discussed it. As for me, such arrangement is quite confusing and disconnected.
2. This paper is more focused on the sketching size. Following the hypergraph setting, the graph structure is very important to the embedding. Currently the author assumes the graph structure is given, but most of times, the structure is unknown. The paper did not further discuss that.
3. The computational cost is efficient on per-iteration level. It still relies on exhaustive search. The overall computational cost can not be effectively measured.
4. Experimental part is locking. The author only conducted general comparison in algorithm 1, but did not show experimental results with baseline methods.

---

> ### Author Response · Authors · 2022-08-02
> **Response to Reviewer QwJS**
>
> We would like to thank the reviewer for the constructive feedback! Our comments to your concerns are as follows:
>
>  Q: The theoretical part of this paper discussed a very general tensor network embedding rooted in the hypergraph presentation. But in the later part, they barely discussed it. As for me, such an arrangement is quite confusing and disconnected.
>
>  A: Thank you for the suggestion! In this paper, we consider the case where the input tensor network DATA has a general hypergraph structure, while the EMBEDDING has a graph structure, rather than a general hypergraph structure. All the analysis, applications as well as experiments follow this. We will address this in the revised version of the paper to avoid confusion.
>
> Q: This paper is more focused on the sketching size. Following the hypergraph setting, the graph structure is very important to the embedding. Currently, the author assumes the graph structure is given, but most of the time, the structure is unknown. The paper did not further discuss that.
>
> A: We consider the following setting in the paper:
>
> The input DATA with its tensor network structure is given,  and the tensor network EMBEDDING structure is chosen based on the data to minimize the sketching cost. Therefore, we didn't assume a given embedding structure in our paper. The embedding graph structure is chosen automatically in Algorithm 1. We also believe that the setting where the input data tensor network structure is known ahead of time is reasonable and common.
>
> Note that we are not considering the altogether different problem of finding a good tensor network decomposition of a single tensor.
>
> Q: The computational cost is efficient on the per-iteration level. It still relies on an exhaustive search. The overall computational cost can not be effectively measured.
>
> A: Thank you for the question! The overall computational costs have two parts: the sketching part (I think this is the part the reviewer mentioned as "per-iteration level"), and the part to decide the embedding structure. Consider the case when there are $N$ contractions in the input data, then the second part has a cost of $O(N)$, which is negligible compared to the first part. We will add this explanation to the revised version of the paper.
>
> As to the comment of "it still relies on exhaustive search": line 263-265 say that "The value of $k(e_j)$ is selected via an exhaustive search over all $|{D}(e_j)|$ contractions". Note that since $|{D}(e_j)|$ is upper-bounded by the number of contractions $N$, this exhaustive search has a cost of $O(N)$, thus is still efficient.
>
> Q: Experimental part is lacking. The author only conducted a general comparison in algorithm 1 but did not show experimental results with baseline methods.
>
> A: We consider multiple baseline algorithms in our experimental section. When comparing sketching tensor train inputs, the tensor train embedding is the baseline and is proposed in [10]. When comparing sketching Kronecker product inputs, the Khatro-Rao product embedding is the baseline and is proposed in [38]. We will explain that Khatro-Rao product embedding is the baseline in the revised version of the paper.
>
> We also add additional experiments that compare CP-ALS and TT rounding using Algorithm 1 with other baseline sketching algorithms for these two applications. Please refer to our response to general questions.
>
> Q: Though most questions are in weakness. I do curious that, is there a good way to assume the hypergraph structure of the tensor network embedding?
>
> A: As mentioned, we consider hypergraph tensor network inputs (e.g., for sketching optimization with CP decomposition), but only consider graph embeddings, for which we can generally bound accuracy.

---

### Official Review · Reviewer_eUJi · 2022-07-11

**Rating:** 6
**Confidence:** 2
**Soundness:** 3 good
**Presentation:** 3 good
**Contribution:** 3 good

**Summary:**

This paper discusses a sketching method for tensor network (TN) structured input data. Unlike previous methods mainly focusing on specific TN structures of the input data, this paper considers more general structures. To alleviate the exponentially large sketching size, the authors impose TN structure on the coefficients. Theoretical guarantees are provided. Finally, the authors also present two applications of the proposed model, including CP-ALS and TT-rounding.

For empirical results, they conducted several synthetic data analyses to justify the theoretical results.


**Questions:**

See above.

**Limitations:**

The authors addressed some limitations. I think one limitation is that they should show some real applications to show the usefulness of the proposed model.

**Strengths And Weaknesses:**

Strengths

1. While previous related works focusing on specific input structure, this work studies a sketching algorithm for general TN structures. This may be useful for more applications. It also improves studies in such fields.

2. The authors also present some theoretical results, which may be beneficial to future work.

3. Based on the proposed algorithms, the authors improve the CP-ALS and TT-rounding algorithms, which are two popular tensor decomposition algorithms used in many applications. So the proposed model may have impacts on potential applications.


Weaknesses

1. The authors conducted several synthetic analyses to justify the theoretical results. However, I think it’s better to also include some real data experiments to show the usefulness of the proposed model.

2. The authors presented two applications of the proposed algorithm, i.e., CP-ALS and TT-rounding. However, they did not conduct experiments on these two applications. I think it is very interesting to show how these algorithms perform in experiments.

3. In this work, the authors aim to study sketching for input data of general TN structure. However, in the experiments, they only used TT and Kronecker structures, both of which are very simple TN structures. Did the authors conduct experiments on more general TNs?

---

> ### Author Response · Authors · 2022-08-02
> **Response to Reviewer eUJi**
>
> We thank the reviewer for the valuable feedback! Our comments to your questions are as follows:
>
> Q: The authors conducted several synthetic analyses to justify the theoretical results. However, I think it’s better to also include some real data experiments to show the usefulness of the proposed model.
>
> A: Thank you for the comments! Please refer to our response to general questions for the additional experiments. For both CP-ALS and TT-rounding, we use a real dataset as the input tensor to justify the efficacy for real cases. These experiments will be added to the revised version of the paper.
>
> Q: The authors presented two applications of the proposed algorithm, i.e., CP-ALS and TT-rounding. However, they did not conduct experiments on these two applications. I think it is very interesting to show how these algorithms perform in experiments.
>
> A: Thank you for the comments! These experiments are presented in our response to general questions.
>
> Q: In this work, the authors aim to study sketching for input data of general TN structure. However, in the experiments, they only used TT and Kronecker structures, both of which are very simple TN structures. Did the authors conduct experiments on more general TNs?
>
> A: In this work, we didn't perform experiments on more general TNs, since this work is the first to present an efficient algorithm to sketch arbitrary TNs, and there is no existing baseline we can compare to. We leave the detailed high-performance implementation of the algorithm for general TNs as future work.

---

### Author Response · Authors · 2022-08-02
**Response to general questions**

We would like to thank all the reviewers for the valuable feedback.
Our comments to the general questions from the reviewers are as follows.

Reviewers eUJi and QwJS asked about additional experiments on applications mentioned in the paper: alternating least squares for CP decomposition and tensor train rounding. We provide additional experiments below to show that our proposed sketching algorithms achieve similar accuracy as state-of-the-art sketching techniques. We will include these results in the revised version of the paper. Note that our proposed sketching algorithms also yields lower or comparable computational cost compared to the baseline sketching techniques, as is already discussed in the paper.

1. For CP-ALS, we perform experiments on a Time-Lapse hyperspectral radiance image (Sérgio MC Nascimento, Kinjiro Amano, and David H Foster. Spatial distributions of local illumination color in natural scenes), which is a 3-D tensor with size $1024 \times 1344 \times 33$. We use this to also justify the usefulness of our method in real cases. We compare the standard CP-ALS, sketched CP-ALS with Algorithm 1 in this work, and sketched CP-ALS with leverage score sampling proposed by Larsen and Kolda in "Practical leverage-based sampling for low-rank tensor decomposition".  We run experiments with 10 ALS iterations, and the output CP decomposition accuracy of these methods under different CP ranks and sketch sizes are as follows:

CP rank: 2  |    5 |   10

sketch size: 25 | 64 |100

CP-ALS: 0.737 | 0.804 | 0.838

CP-ALS with out sketching method: 0.737 | 0.770 | 0.801

CP-ALS with leverage score sampling by  Larsen and Kolda: 0.739 | 0.773 | 0.789

As can be seen from the results above, the CP-ALS output accuracy with our method has comparable accuracy with the sketched CP-ALS algorithm with leverage score sampling. As is analyzed in 347-355 in the paper, our algorithm also has better complexity compared to the baseline algorithm, especially when the CP rank is low and the tensor dimension is large.

2. For tensor train rounding, we also perform experiments on data from the Time-Lapse hyperspectral radiance image dataset. We use 9 images from the dataset, and reshaping the input data to an order 6 tensor with size $9 \times 32 \times 32 \times 28 \times  48 \times 33$. We then use the TensorLy library to truncate the input tensor to a tensor train with a rank of 30 and then benchmark the accuracy of different methods on top of this tensor train.

   With different TT rounding rank thresholds and different sketch size, the truncated tensor train accuracies are presented as follows:

   TT rounding rank: 1 | 4 | 11 | 20

   sketch size: 4 | 9 | 16 | 25

   TT-SVD: 0.734 | 0.862 | 0.944 | 0.981

   sketching with Algorithm 1:  0.527 | 0.761 | 0.866 | 0.948

   sketching with MPS embedding: 0.573 | 0.757 | 0.882 | 0.951

 As can be seen from the results above, sketching with our method (Algorithm 1) has comparable accuracy with the baseline algorithm (sketching with MPS embedding). In addition,
 As is analyzed in 362-379 in the paper, our algorithm also has similar complexity compared to sketching with MPS embedding,
 meaning that both methods have similar accuracy and computational cost performance in TT rounding.

---

### Meta-Review · Area_Chair_ga7E · 2022-09-07

**Recommendation:** Accept
**Confidence:** Less certain

**Metareview:**

Summary:

The major strength is that the sketch size is polynomial in the number of modes for tensor train. This was not known in previous work, for example in the paper by Rakhshan and Rabusseau https://arxiv.org/abs/2003.05101 which is reference 38 and gets a sketch size which is exponential in the number of modes. There is concurrent work that also gets this here: https://arxiv.org/abs/2207.07417

Theorem 4.3 seems like it could be of independent interest. It shows that under reasonable assumptions, assuming the contraction order of the data tensor is fixed, if the embedding is a tree, then this gets the optimal running time - more complex embeddings are not needed (even if the data tensor has cycles). Section 4 is focused on giving an algorithm such that, given a data tensor and an embedding, and a fixed contraction order for the data tensor, applies the embedding to the data tensor with the smallest possible asymptotic running time. Not sure if previous work has studied this question, however.

One weakness is that the works doesn’t seem to compare to previous work related to Section 4, or running time for subspace embeddings for tensor networks. Their CP decomposition algorithm is also incomparable to the other CP decomposition algorithms they mention, since their per-iteration running time for ALS has a 1/eps^5 term. Perhaps if s (the size of each dimension of the tensor) is much bigger than N (the number of modes) or R (the rank), then the second term in the running time should dominate, meaning that the algorithm in this paper would still be significantly better than Recursive LSS (by a factor of NR as they mention in page 8), so this might not be a significant weakness.

Evaluation:

Based on the reviews and my understanding, I think this meets the bar for NeurIPS. The sketch size for tensor train and the application to CP seem useful, and tensor train rounding seems to be a strong motivation. The significance of Section 4 is not completely clear, but the other results seem to be enough by themselves.


**Award:**

No

---

### Decision · Program_Chairs · 2022-09-14

Accept